# On the Effect of Batch Size
# in Byzantine-Robust Distributed Learning

**Yi-Rui Yang**      **Chang-Wei Shi**      **Wu-Jun Li**[*]
National Key Laboratory for Novel Software Technology,
Department of Computer Science and Technology,
Nanjing University, Nanjing, China
`{yangyr, shicw}@smail.nju.edu.cn, liwujun@nju.edu.cn`

## Abstract

Byzantine-robust distributed learning (BRDL), in which computing devices are likely to behave abnormally due to accidental failures or malicious attacks, has recently become a hot research topic. However, even in the independent and identically distributed (i.i.d.) case, existing BRDL methods will suffer a significant drop on model accuracy due to the large variance of stochastic gradients. Increasing batch size is a simple yet effective way to reduce the variance. However, when the total number of gradient computation is fixed, a too-large batch size will lead to a too-small iteration number (update number), which may also degrade the model accuracy. In view of this challenge, we mainly study the effect of batch size when the total number of gradient computation is fixed in this work. In particular, we show that when the total number of gradient computation is fixed, the optimal batch size corresponding to the tightest theoretical upper bound in BRDL increases with the fraction of Byzantine workers. Therefore, compared to the case without attacks, a larger batch size is preferred when under Byzantine attacks. Motivated by the theoretical finding, we propose a novel method called Byzantine-robust stochastic gradient descent with normalized momentum (ByzSGDnm) in order to further increase model accuracy in BRDL. We theoretically prove the convergence of ByzSGDnm for general non-convex cases under Byzantine attacks. Empirical results show that when under Byzantine attacks, using a relatively large batch size can significantly increase the model accuracy, which is consistent with our theoretical results. Moreover, ByzSGDnm can achieve higher model accuracy than existing BRDL methods when under deliberately crafted attacks. In addition, we empirically show that increasing batch size has the bonus of training acceleration.

## 1  Introduction

Distributed learning has attracted much attention (Haddadpour et al., 2019; Jaggi et al., 2014; Lee et al., 2017; Lian et al., 2017; Ma et al., 2015; Shamir et al., 2014; Sun et al., 2018; Yang, 2013; Yu et al., 2019a;b; Zhao et al., 2017; 2018; Zhou et al., 2018; Zinkevich et al., 2010) for years due to its wide application. In traditional distributed learning, it is typically assumed that there is no failure or attack. However, in some real-world applications such as edge-computing (Shi et al., 2016) and federated learning (McMahan & Ramage, 2017), the service provider (also known as the server) usually has weak control over computing nodes (also known as workers). In these cases, various software and hardware failures may happen on workers (Xie et al., 2019). Worse even, some workers may get hacked by a malicious third party and intentionally send wrong information to foil the distributed learning process (Kairouz et al., 2021). The workers under failure or attack are also called Byzantine workers. Distributed learning with the existence of Byzantine workers, which is also known as Byzantine-robust distributed learning (BRDL), has recently become a hot research topic (Bernstein et al., 2019; Bulusu et al., 2021; Chen et al., 2018; Damaskinos et al., 2018; Diakonikolas et al., 2017; Diakonikolas & Kane, 2019; Konstantinidis & Ramamoorthy, 2021; Lamport et al., 2019; Rajput et al., 2019; Sohn et al., 2020; Wu et al., 2020; Yang & Li, 2021; 2023; Yang et al., 2020; Yin et al., 2019).

---

[*]Corresponding author.

A typical way to obtain Byzantine robustness is to substitute the mean aggregator with robust aggregators such as Krum (Blanchard et al., 2017), geometric median (Chen et al., 2017), coordinate-wise median (Yin et al., 2018), centered clipping (Karimireddy et al., 2021), and so on. However, when there are Byzantine workers, even if robust aggregators are used, it is inevitable that an aggregation error will be introduced, which is the difference between the aggregated result and the true mean value. Furthermore, even in the independent and identically distributed (i.i.d.) cases, the aggregation error could be large due to the large variance of stochastic gradients (Karimireddy et al., 2021) which are typical values sent from workers to the server for parameter updating. The large aggregation error would make BRDL methods fail (Xie et al., 2020).

It has been shown in existing works that the variance of the values from non-Byzantine workers can be reduced by using local momentum on workers (Allen-Zhu et al., 2020; El-Mhamdi et al., 2021; Farhadkhani et al., 2022; Karimireddy et al., 2021). However, as the empirical results in our work will show, even if local momentum has been used, existing BRDL methods will suffer a significant drop on model accuracy when under attacks. Therefore, more sophisticated techniques are required to further reduce the variance of stochastic gradients.

Increasing batch size is a simple yet effective way to reduce the variance. However, when the total number of gradient computation is fixed, a too-large batch size will lead to a too-small iteration number (update number), which may also degrade the model accuracy (Goyal et al., 2017; Hoffer et al., 2017; Keskar et al., 2017; You et al., 2020; Zhao et al., 2020; 2023). In view of this challenge, we mainly study the effect of batch size in i.i.d. cases when the total number of gradient computation is fixed. The main contributions of this work are listed as follows:

- We show that when the total number of gradient computation is fixed, the optimal batch size corresponding to the tightest theoretical upper bound in BRDL increases with the fraction of Byzantine workers.
- Motivated by the theoretical finding, we propose a novel method called Byzantine-robust stochastic gradient descent with normalized momentum (ByzSGDnm) in order to further increase model accuracy in BRDL.
- We theoretically prove the convergence of ByzSGDnm for non-convex cases under attacks.
- We empirically show that when under Byzantine attacks, compared to the cases of small batch size, setting a relatively large batch size can significantly increase the model accuracy. Moreover, ByzSGDnm can achieve higher model accuracy than existing BRDL methods when under deliberately crafted attacks.
- In addition, increasing batch size has the bonus of training acceleration, which is verified by our empirical results.

## 2 PRELIMINARY

In this paper, we mainly focus on the following optimization problem:

$$\min_{\mathbf{w} \in \mathbb{R}^d} F(\mathbf{w}) = \mathbb{E}_{\xi \sim \mathcal{D}}[f(\mathbf{w}, \xi)], \tag{1}$$

where $\mathbf{w} \in \mathbb{R}^d$ is the model parameter and $\mathcal{D}$ is the distribution of training data. In addition, we mainly focus on the widely-used parameter-server (PS) framework in this work, where there are $m$ computing nodes (workers) that collaborate to train the learning model under the coordination of a central server. Each worker can independently draw samples $\xi$ from data distribution $\mathcal{D}$. That is to say, we focus on the i.i.d. cases in this paper. Moreover, among the $m$ workers, a fraction of $\delta$ workers are Byzantine, which may behave abnormally and send arbitrary values to the server due to accidental failure or malicious attacks. The other workers, which are called non-Byzantine workers, will faithfully conduct the training algorithm without any fault. Formally, we use $\mathcal{G} \subseteq \{1, 2, \ldots, m\}$ to denote the index set of non-Byzantine workers where $|\mathcal{G}| = (1 - \delta)m$. The server has no access to any training data and does not know which workers are Byzantine. In this work, we mainly consider the loss functions that satisfy the following three assumptions, which are quite common in distributed learning. For simplicity, we use the notation $\|\cdot\|$ to denote the Euclidean norm of a vector.

**Assumption 1** (Bounded variance). *There exists $\sigma \geq 0$, such that $\mathbb{E}_{\xi \sim \mathcal{D}} \|\nabla f(\mathbf{w}, \xi) - \nabla F(\mathbf{w})\|^2 \leq \sigma^2$ for all $\mathbf{w} \in \mathbb{R}^d$.*

**Assumption 2** (Lower bound of $F(\cdot)$). *There exists $F^* \in \mathbb{R}$ such that $F(\mathbf{w}) \geq F^*$ for all $\mathbf{w} \in \mathbb{R}^d$.*

**Assumption 3** ($L$-smoothness). *The loss function $F(\cdot)$ is differentiable everywhere on $\mathbb{R}^d$. Moreover, $\|\nabla F(\mathbf{w}) - \nabla F(\mathbf{w}')\| \leq L\|\mathbf{w} - \mathbf{w}'\|$ for all $\mathbf{w}, \mathbf{w}' \in \mathbb{R}^d$.*

A typical and widely-used algorithm to solve the optimization problem (1) with potential Byzantine workers is Byzantine-robust stochastic gradient descent with momentum (ByzSGDm) (Farhadkhani et al., 2022; Karimireddy et al., 2021). Compared with vanilla stochastic gradient descent with momentum (SGDm), the main difference in ByzSGDm is that the mean aggregator on the server is substituted by a robust aggregator. Specifically, in ByzSGDm, the server updates the model parameter at the $t$-th iteration by computing

$$\mathbf{w}_{t+1} = \mathbf{w}_t - \eta_t \cdot \mathbf{Agg}(\mathbf{u}_t^{(1)}, \ldots, \mathbf{u}_t^{(m)}),$$

where $\eta_t$ is the learning rate and $\mathbf{Agg}(\cdot)$ is a robust aggregator. Local momentum $\mathbf{u}_t^{(k)}$ is received from the $k$-th worker ($k = 1, 2, \ldots, m$). For each non-Byzantine worker $k \in \mathcal{G}$,

$$\mathbf{u}_t^{(k)} = \begin{cases} \mathbf{g}_0^{(k)}, & t = 0; \\ \beta\mathbf{u}_{t-1}^{(k)} + (1-\beta)\mathbf{g}_t^{(k)}, & t > 0, \end{cases}$$

where $\beta$ is the momentum hyper-parameter and $\mathbf{g}_t^{(k)} = \frac{1}{B}\sum_{b=1}^{B}\nabla f(\mathbf{w}_t, \xi_t^{(k,b)})$ is the mean value of a mini-batch of stochastic gradients with size $B$. For each Byzantine worker $k \in [m] \setminus \mathcal{G}$, $\mathbf{u}_t^{(k)}$ can be an arbitrary value. For space saving, more details about ByzSGDm are moved to Algorithm 2 and Algorithm 3 in Appendix A.

For a 'good' aggregator, the aggregated result $\mathbf{Agg}(\mathbf{u}_t^{(1)}, \ldots, \mathbf{u}_t^{(m)})$ should be close to the true mean of the momentums on non-Byzantine workers, which can be written as $\frac{1}{|\mathcal{G}|}\sum_{k \in \mathcal{G}}\mathbf{u}_t^{(k)}$. To quantitatively measure a robust aggregator, the definition of $(\delta_{\max}, c)$-robust aggregator has been proposed in existing works (Karimireddy et al., 2021), which we present in Definition 1 below.

**Definition 1** (($\delta_{\max}, c$)-robust aggregator (Karimireddy et al., 2021)). *Let $0 \leq \delta_{\max} < \frac{1}{2}$ and $c \geq 0$. Random vectors $\mathbf{x}_1, \ldots, \mathbf{x}_m \in \mathbb{R}^d$ satisfy that $\mathbb{E}\|\mathbf{x}_k - \mathbf{x}_{k'}\|^2 \leq \rho^2$ for all fixed $k, k' \in \mathcal{G}$, where $\mathcal{G} \subseteq \{1, \ldots, m\}$ and $|\mathcal{G}| = (1-\delta)m$. An aggregator $\mathbf{Agg}(\cdot)$ is called a ($\delta_{\max}, c$)-robust aggregator if we always have that*

$$\mathbb{E}\|\mathbf{e}\|^2 \leq c\delta\rho^2,$$

*when $\delta \leq \delta_{\max}$. Here, $\mathbf{e} = \mathbf{Agg}(\mathbf{x}_1, \ldots, \mathbf{x}_m) - \frac{1}{|\mathcal{G}|}\sum_{k \in \mathcal{G}}\mathbf{x}_k$ is called the aggregation error.*

In addition, it has been proved that for any potential robust aggregator, there is inevitably an aggregation error of $\Omega(\delta\rho^2)$ in the worst case (Karimireddy et al., 2021). It has also been proved that some existing aggregators such as centered clipping (Karimireddy et al., 2021) satisfy Definition 1.

## 3 METHODOLOGY

### 3.1 EFFECT OF BATCH SIZE ON CONVERGENCE

As shown in existing works on Byzantine-robust distributed learning (Blanchard et al., 2017; Chen et al., 2017; Li et al., 2019; Yin et al., 2018), even if robust aggregators have been used, there is typically a drop on model accuracy under Byzantine attacks due to the aggregation error. Therefore, we attempt to alleviate the drop on model accuracy by reducing the aggregation error.

According to Definition 1, there are three variables related to the upper bound of aggregation error. The fraction of Byzantine workers $\delta$ is determined by the problem, which can hardly be reduced. The constant $c$ is mainly related to the specific robust aggregator. There have been many works (Blanchard et al., 2017; Chen et al., 2017; Karimireddy et al., 2021; Li et al., 2019; Yin et al., 2018) that propose various robust aggregators. In this work, we mainly attempt to reduce $\rho$. Moreover, we focus on the i.i.d. setting in this work. Since $\mathbb{E}[\mathbf{x}_k] = \mathbb{E}[\mathbf{x}_{k'}]$ in this case, according to Assumption 1, we have

$$\mathbb{E}\|\mathbf{x}_k - \mathbf{x}_{k'}\|^2 = \mathbb{E}\|(\mathbf{x}_k - \mathbb{E}[\mathbf{x}_k]) - (\mathbf{x}_{k'} - \mathbb{E}[\mathbf{x}_{k'}])\|^2 = \mathbb{E}\|\mathbf{x}_k - \mathbb{E}[\mathbf{x}_k]\|^2 + \mathbb{E}\|\mathbf{x}_{k'} - \mathbb{E}[\mathbf{x}_{k'}]\|^2 \leq 2\sigma^2,$$
(2)

which implies that $\rho^2 \leq 2\sigma^2$ in i.i.d. cases under Assumption 1. Therefore, we can reduce $\rho$ by reducing the variance $\sigma^2$ in i.i.d. cases. A simple but effective way to reduce the variance is increasing the batch size on each worker, which is denoted by $B$ in this paper. For simplicity, we assume that all workers adopt the same batch size in this work. Compared to the case with batch size 1, the variance of stochastic gradients will be reduced to $1/B$ of the original if the batch size is set to $B$. However, to make the total number of gradient computation unchanged, the total iteration number will be reduced to $1/B$ of the original, leading to fewer times of model updating. Formally, we use $\mathcal{C} = TBm(1-\delta)$ to denote the total number of gradient computation on non-Byzantine workers, where $T$ is the total iteration number. Thus, we have $T = \frac{\mathcal{C}}{Bm(1-\delta)}$. It implies that a larger batch size $B$ will lead to a smaller total iteration number $T$ when the total number of gradient computation $\mathcal{C}$ is fixed. In many BRDL applications with deep learning models, $\mathcal{C}$ can be used to approximately evaluate the computation cost since the computation cost of robust aggregation and model updating is negligible compared to that of gradient computation.

We first recall the convergence of ByzSGDm, which has been adequately studied in existing works (Karimireddy et al., 2021). We restate the convergence results of ByzSGDm in Theorem 1 below. For space saving, the details of how Theorem 1 is obtained from existing results (Karimireddy et al., 2021) are presented in Appendix B .

**Theorem 1** (Convergence of ByzSGDm (Karimireddy et al., 2021)). *Suppose that $F(\mathbf{w}_0) - F^* \leq F_0$. Under Assumptions 1, 2 and 3, when $\mathbf{Agg}(\cdot)$ is $(\delta_{\max}, c)$-robust and $\delta \leq \delta_{\max}$, setting $\eta_t = \eta = \min\left(\sqrt{\frac{F_0 + \frac{5c\delta\sigma^2}{16BL}}{\frac{20LT\sigma^2}{B}\left(\frac{2}{m} + c\delta\right)}}, \frac{1}{8L}\right)$ and $1 - \beta = 8L\eta$, we have the following result for ByzSGDm:*

$$\frac{1}{T} \sum_{t=0}^{T-1} \mathbb{E}\|\nabla F(\mathbf{w}_t)\|^2 \leq 16\sqrt{\frac{\sigma^2(1+c\delta m)}{TBm}} \left(\sqrt{10LF_0} + \sqrt{\frac{3c\delta\sigma^2}{B}}\right) + \frac{32LF_0}{T} + \frac{20\sigma^2(1+c\delta m)}{TBm}. \tag{3}$$

When $\mathcal{C}$ is fixed, inequality (3) can be re-written as $\frac{1}{T} \sum_{t=0}^{T-1} \mathbb{E}\|\nabla F(\mathbf{w}_t)\|^2 \leq \mathcal{U}(B)$ since $T = \frac{\mathcal{C}}{Bm(1-\delta)}$, where $\mathcal{U}(B)$ is a real-valued function with respect to batch size $B$. Specifically,

$$\mathcal{U}(B) = 16\sqrt{\frac{\sigma^2(1+c\delta m)(1-\delta)}{\mathcal{C}}} \left(\sqrt{10LF_0} + \sqrt{\frac{3c\delta\sigma^2}{B}}\right) + \frac{32LF_0 Bm(1-\delta)}{\mathcal{C}}$$
$$+ \frac{20\sigma^2(1+c\delta m)(1-\delta)}{\mathcal{C}}.$$

Please note that $\mathcal{U}(B)$ is originally defined on the set of positive integers $\mathbb{N}^*$ since $B$ denotes the batch size. We here extend the definition of $\mathcal{U}(B)$ to $B \in (0, +\infty)$ for simplicity. The results will be interpreted back to $B \in \mathbb{N}^*$ at the end of our analysis. Then we attempt to find the optimal batch size $B^*$ that minimizes the theoretical upper bound $\mathcal{U}(B)$ when $\mathcal{C} = TBm(1-\delta)$ is fixed. Formally, $B^*$ is defined by the following optimization problem:

$$B^* = \underset{B \in (0, +\infty)}{\arg\min} \ \mathcal{U}(B).$$

We present Proposition 1 below, which provides an explicit expression of $B^*$. Please refer to Appendix B for the proof details.

**Proposition 1.** *$\mathcal{U}(B)$ is strictly convex on $(0, +\infty)$. Moreover, when $\delta > 0$, we have*

$$B^* = \left(\frac{3}{16L^2(F_0)^2 m}\right)^{\frac{1}{3}} \left(\frac{c\delta(1+c\delta m)}{m(1-\delta)}\right)^{\frac{1}{3}} \sigma^{\frac{4}{3}} \mathcal{C}^{\frac{1}{3}}, \tag{4}$$

*and*

$$\mathcal{U}(B^*) = \frac{16\sqrt{10LF_0(1+c\delta m)(1-\delta)}\sigma}{\mathcal{C}^{\frac{1}{2}}} + \frac{24\left[12c\delta(1+c\delta m)(1-\delta)^2 LF_0 m\right]^{\frac{1}{3}} \sigma^{\frac{4}{3}}}{\mathcal{C}^{\frac{2}{3}}}$$
$$+ \frac{20(1+c\delta m)(1-\delta)\sigma^2}{\mathcal{C}}.$$

---

**Algorithm 1** Byzantine-Robust SGD with Normalized Momentum (ByzSGDnm)

---

**Input:** initial model parameter $\mathbf{w}_0$, worker number $m$, iteration number $T$, learning rates $\{\eta_t\}_{t=0}^{T-1}$, batch size $B$, momentum hyper-parameter $\beta \in [0, 1)$, robust aggregator $\mathbf{Agg}(\cdot)$;

**for** $t = 0$ **to** $T - 1$ **do**

    Broadcast $\mathbf{w}_t$ to all workers;

    **on worker** $k \in \{1, \ldots, m\}$ **in parallel do**

        Receive $\mathbf{w}_t$ from the server;

        Independently draw $B$ samples $\xi_t^{(k,1)}, \ldots, \xi_t^{(k,B)}$ from distribution $\mathcal{D}$;

        Compute $\mathbf{g}_t^{(k)} = \frac{1}{B} \sum_{b=1}^{B} \nabla f(\mathbf{w}_t, \xi_t^{(k,b)})$;

        Update local momentum $\mathbf{u}_t^{(k)} = \begin{cases} \mathbf{g}_0^{(k)}, & t = 0; \\ \beta \mathbf{u}_{t-1}^{(k)} + (1 - \beta) \mathbf{g}_t^{(k)}, & t > 0; \end{cases}$

        Send $\mathbf{u}_t^{(k)}$ to the server (Byzantine workers may send arbitrary values at this step);

    **end on worker**

    Receive $\{\mathbf{u}_t^{(k)}\}_{k=1}^{m}$ from the $m$ workers, and compute $\mathbf{u}_t = \mathbf{Agg}(\mathbf{u}_t^{(1)}, \ldots, \mathbf{u}_t^{(m)})$;

    Update model parameter with normalized momentum: $\mathbf{w}_{t+1} = \mathbf{w}_t - \eta_t \frac{\mathbf{u}_t}{\|\mathbf{u}_t\|}$;

**end for**

Output model parameter $\mathbf{w}_T$.

---

Please note that $\mathcal{U}(B)$ has no more than one global minimizer due to the strict convexity. Thus, $B^*$ is well-defined when $\delta > 0$. Furthermore, the term $\left( \frac{c\delta(1+c\delta m)}{m(1-\delta)} \right)^{\frac{1}{3}}$ in (4) is monotonically increasing with respect to $\delta$. It implies that when the total number of gradient computation on non-Byzantine workers $\mathcal{C} = TBm(1 - \delta)$ is fixed, $B^*$ will increase as the fraction of Byzantine workers $\delta$ increases. Then, we interpret the above results back to $B \in \mathbb{N}^*$. Due to the strict convexity, $\mathcal{U}(B)$ is monotonically decreasing when $B \in (0, B^*)$ and monotonically increasing when $B \in (B^*, +\infty)$. Thus, the optimal integer batch size that minimizes $\mathcal{U}(B)$ equals either $\lfloor B^* \rfloor$ or $\lfloor B^* \rfloor + 1$, which also increases with $\delta$. The notation $\lfloor B^* \rfloor$ represents the largest integer that is not larger than $B^*$. In addition, the conclusion will be further supported by the empirical results in Section 5.

Meanwhile, although $B^* \to 0$ as $\delta \to 0^+$, it should not be interpreted as recommending a batch size that is close to 0 when there is no attack. In fact, since $\mathcal{C}$ is fixed, a too-small batch size $B$ implies a too-large iteration number $T$, which will lead to a large communication cost. Moreover, the computation power of some devices (e.g., GPUs) will not be effectively utilized when $B$ is too small. Thus, the setting of $B$ is a trade-off between model accuracy and running time when $\delta = 0$, which has been studied for years (Goyal et al., 2017; You et al., 2020; Zhao et al., 2020; 2023). In addition, please note that although a smaller $\mathcal{U}(B)$ does not necessarily ensure a better empirical performance given the complexity and variety in real-world applications, it provides a better worst-case guarantee.

## 3.2 BYZANTINE-ROBUST SGD WITH NORMALIZED MOMENTUM

Proposition 1 shows that the optimal batch size $B^*$ that minimizes $\mathcal{U}(B)$ increases with the fraction of Byzantine workers. Hence, a relatively large batch size is preferred when under Byzantine attacks. In existing works on traditional large-batch training without attacks (Goyal et al., 2017; Hoffer et al., 2017; Keskar et al., 2017; Zhao et al., 2020; 2023), the normalization technique is widely used to increase model accuracy. Motivated by this, we propose a novel method called Byzantine-robust stochastic gradient descent with normalized momentum (ByzSGDnm), by introducing a simple normalization operation. Specifically, in ByzSGDnm, the model parameters are updated by:

$$\mathbf{w}_{t+1} = \mathbf{w}_t - \eta_t \cdot \frac{\mathbf{Agg}(\mathbf{u}_t^{(1)}, \ldots, \mathbf{u}_t^{(m)})}{\|\mathbf{Agg}(\mathbf{u}_t^{(1)}, \ldots, \mathbf{u}_t^{(m)})\|}.$$

The details of ByzSGDnm are illustrated in Algorithm 1. Please note that there are some existing methods using layer-wise normalization (You et al., 2020). However, these methods might suffer from degradation of model accuracy without additional training tricks such as warm-up (Zhao et al., 2023). Furthermore, as shown in (Zhao et al., 2023), the layer-wise (block-wise) normalization might slow down the convergence rate. Hence, we follow the way of performing normalization on the whole momentum (Zhao et al., 2020; 2023; Cutkosky & Mehta, 2020).

Moreover, please note that the purpose of traditional large-batch training (Goyal et al., 2017; You et al., 2020; Zhao et al., 2020; 2023) is mainly to accelerate the training process by reducing communication cost and utilizing the computation power more effectively. However, in this work, the main purpose of increasing batch size and using momentum normalization is to enhance the Byzantine robustness and increase the model accuracy under Byzantine attacks. The acceleration effect of adopting large batch size is viewed as a bonus in this work. Please refer to Section 5 for the empirical results about the wall-clock time of ByzSGDm and ByzSGDnm with different batch size.

## 4  CONVERGENCE

In this section, we theoretically analyze the convergence of ByzSGDnm under Assumptions 1, 2 and 3. The assumptions are common in distributed learning. For space saving, we only present the main results here. Please refer to Appendix B for the proof details.

**Theorem 2.** *Suppose that $F(\mathbf{w}_0) - F^* \leq F_0$ and let $\alpha = 1 - \beta$. Under Assumptions 1, 2 and 3, when $\mathbf{Agg}(\cdot)$ is $(\delta_{\max}, c)$-robust, $\delta \leq \delta_{\max}$ and $\eta_t = \eta$, we have the following result for ByzSGDnm:*

$$\frac{1}{T}\sum_{t=0}^{T-1}\mathbb{E}\|\nabla F(\mathbf{w}_t)\| \leq \frac{2F_0}{\eta T} + \frac{10\eta L}{\alpha} + \frac{9\sqrt{2cm\delta(1-\delta)}+9}{\sqrt{Bm(1-\delta)}}\left(\frac{1}{\alpha T} + \sqrt{\alpha}\right)\sigma.$$

Finally, we show that when the learning rate $\eta$ and the momentum hyper-parameter $\beta = 1 - \alpha$ are properly set, ByzSGDnm can achieve the convergence order of $O\left(\frac{1}{T^{\frac{1}{4}}}\right)$ by Proposition 2 below.

**Proposition 2.** *Under Assumptions 1, 2 and 3, when $\mathbf{Agg}(\cdot)$ is $(\delta_{\max}, c)$-robust and $\delta \leq \delta_{\max}$, setting $1 - \beta = \alpha = \min\left(\frac{\sqrt{80LF_0Bm(1-\delta)}}{\left[9\sqrt{2cm\delta(1-\delta)}+9\right]\sigma\sqrt{T}}, 1\right)$ and $\eta_t = \eta = \sqrt{\frac{\alpha F_0}{5LT}}$, we have that*

$$\frac{1}{T}\sum_{t=0}^{T-1}\mathbb{E}\|\nabla F(\mathbf{w}_t)\| \leq 6\left[\sqrt{2cm\delta(1-\delta)}+1\right]^{\frac{1}{2}}\left(\frac{5LF_0\sigma^2}{TBm(1-\delta)}\right)^{\frac{1}{4}} + 12\sqrt{\frac{5LF_0}{T}}$$

$$+ \frac{27\left[\sqrt{2cm\delta(1-\delta)}+1\right]^{\frac{3}{2}}}{4\sqrt{5TB^2m^2(1-\delta)^2LF_0}}\sigma^2. \quad (5)$$

*Moreover, when $\mathcal{C} = TBm(1-\delta)$ is fixed, the optimal batch size $\tilde{B}^*$ that minimizes the right-hand side of (5) is $\tilde{B}^* = \frac{9\left[\sqrt{2cm\delta(1-\delta)}+1\right]^{\frac{3}{2}}\sigma^2}{80m(1-\delta)LF_0}$. In this case ($B = \tilde{B}^*$), we have:*

$$\frac{1}{T}\sum_{t=0}^{T-1}\mathbb{E}\|\nabla F(\mathbf{w}_t)\| \leq \frac{6\left[\sqrt{2cm\delta(1-\delta)}+1\right]^{\frac{1}{2}}\left(5LF_0\sigma^2\right)^{\frac{1}{4}}}{\mathcal{C}^{\frac{1}{4}}} + \frac{18\left[\sqrt{2cm\delta(1-\delta)}+1\right]^{\frac{3}{4}}\sigma}{\mathcal{C}^{\frac{1}{2}}}.$$

Inequality (5) illustrates that after $T$ iterations, ByzSGDnm can guarantee that

$$\min_{t=0,\dots,T-1}\mathbb{E}\|\nabla F(\mathbf{w}_t)\| \leq O\left(\frac{(LF_0)^{\frac{1}{4}}\sqrt{\sigma}}{T^{\frac{1}{4}}} + \frac{1}{T^{\frac{1}{2}}}\right).$$

Therefore, ByzSGDnm has the same convergence order as vanilla SGD with normalized momentum (Cutkosky & Mehta, 2020) without attacks. The extra factor $[\sqrt{2cm\delta(1-\delta)}+1]^{\frac{1}{2}}/(1-\delta)^{\frac{1}{4}}$ in the right-hand side (RHS) of (5) is due to the existence of Byzantine workers and increases with $\delta$. The extra factor vanishes (equals 1) when there is no Byzantine worker ($\delta = 0$). Moreover, it has been shown in existing works (Arjevani et al., 2023; Cutkosky & Mehta, 2020) that under Assumptions 1, 2 and 3, the convergence order $O(1/T^{\frac{1}{4}})$ is optimal for SGD. Byz-VR-MARINA (Gorbunov et al., 2023) achieves a better convergence order by intermittently using full gradients. However, full gradients are computationally expensive, especially in real-world applications with a large number of training instances. We detailedly compare ByzSGDnm with Byz-VR-MARINA in Appendix D. The empirical results show that ByzSGDnm significantly outperforms Byz-VR-MARINA. In addition, $\tilde{B}^*$ also increases with $\delta$ since both $\delta(1-\delta)$ and $\frac{1}{1-\delta}$ increase with $\delta$ when $\delta \in [0, \frac{1}{2})$.

The analysis in this paper is based on the definition of $(\delta_{\max}, c)$-robust aggregator (Definition 1). There is also another criterion of robust aggregators in existing works called the $(f, \kappa)$-robustness (Allouah et al., 2023). Similar results can also be obtained under the $(f, \kappa)$-robustness. Please refer to Appendix C for more details.

## 5 EXPERIMENT

**Task and platform.** In this section, we will empirically test the performance of ByzSGDm and ByzSGDnm on image classification tasks. Each algorithm will be used to train a ResNet-20 (He et al., 2016) deep learning model on CIFAR-10 dataset (Krizhevsky et al., 2009). All the experiments presented in this work are conducted on a distributed platform with 9 dockers. Each docker is bound to an NVIDIA TITAN Xp GPU. One docker is chosen as the server while the other 8 dockers are chosen as workers. The training instances are randomly and equally distributed to the workers.

**Experimental settings.** In existing works (Allouah et al., 2023; Karimireddy et al., 2021; 2022) on BRDL, the batch size is typically set to 32 or 50 on the CIFAR-10 dataset. Therefore, We set ByzSGDm (Karimireddy et al., 2021) with batch size 32 as the baseline, and compare the performance of ByzSGDm with different batch size (ranging from 64 to 1024) to the baseline under ALIE attack (Baruch et al., 2019). In our experiments, we use four widely-used robust aggregators Krum (KR) (Blanchard et al., 2017), geometric median (GM) (Chen et al., 2017), coordinate-wise median (CM) (Yin et al., 2018) and centered clipping (CC) (Karimireddy et al., 2021) for ByzSGDm. Moreover, we set the clipping radius to $0.1$ for CC. We train the model for 160 epochs with cosine annealing learning rates (Loshchilov & Hutter, 2017). Specifically, the learning rate at the $i$-th epoch will be $\eta_i = \frac{\eta_0}{2}(1 + \cos(\frac{i}{160}\pi))$ for $i = 0, 1, \ldots, 159$. The initial learning rate $\eta_0$ is selected from $\{0.1, 0.2, 0.5, 1.0, 2.0, 5.0, 10.0, 20.0\}$, and the best final top-1 test accuracy is used as the final metrics. The momentum hyper-parameter $\beta$ is set to $0.9$. Please note that the total number of gradient computation on non-Byzantine workers $\mathcal{C}$ is independent of batch size. Specifically, $\mathcal{C} = 160 \times 50000 \times (1 - \delta)$ since we train the model for 160 epochs with 50000 training instances.

**Evaluation on the effect of batch size.** We first evaluate the performance of ByzSGDm with different batch size when the fraction of Byzantine workers $\delta$ is 0 (no attack), $\frac{1}{8}$ and $\frac{3}{8}$, respectively. As the results in Table 1 and Table 2 show, the batch size corresponding to the best top-1 accuracy increases with $\delta$, which is consistent with our theoretical results. Moreover, when $\delta = \frac{3}{8}$, using a relatively large batch size greatly increases the test accuracy. Meanwhile, the test accuracy decreases with the batch size when there is no attack ($\delta = 0$), which is consistent with existing works (Goyal et al., 2017; Hoffer et al., 2017; Keskar et al., 2017; You et al., 2020; Zhao et al., 2020; 2023).

Table 1: The final top-1 test accuracy of ByzSGDm with various batch size under ALIE attack when Krum (KM) and geometric median (GM) are used as the robust aggregator

| Batch size | ByzSGDm with KR | | |
|---|---|---|---|
| | $\delta = 0$ | $\delta = 1/8$ | $\delta = 3/8$ |
| 32×8 (baseline) | **91.08%** | 55.84% | 38.55% |
| 64×8 | 89.98% (-1.10%) | 63.22% (+7.38%) | 54.15% (+15.60 %) |
| 128×8 | 89.71% (-1.37%) | 75.06% (+19.22%) | 55.98% (+17.43%) |
| 256×8 | 89.15% (-1.93%) | 84.47% (+28.63%) | 59.28% (+20.73%) |
| 512×8 | 86.15% (-4.93%) | **85.68% (+29.84%)** | 83.42% (+44.87%) |
| 1024×8 | 84.97% (-6.11%) | 83.48% (+27.64%) | **83.45% (+44.90%)** |

| Batch size | ByzSGDm with GM | | |
|---|---|---|---|
| | $\delta = 0$ | $\delta = 1/8$ | $\delta = 3/8$ |
| 32×8 (baseline) | **92.02%** | 83.81% | 63.11% |
| 64×8 | 91.50% (-0.52%) | 87.92% (+4.11%) | 70.88% (+7.77%) |
| 128×8 | 90.85% (-1.17%) | **89.68% (+5.87%)** | 82.08% (+18.97%) |
| 256×8 | 89.26% (-2.76%) | 87.99% (+4.18%) | **87.62% (+24.51%)** |
| 512×8 | 88.21% (-3.81%) | 87.70% (+3.89%) | 86.95% (+23.84%) |
| 1024×8 | 86.52% (-5.50%) | 85.94% (+2.13%) | 84.75% (+21.64%) |

Table 2: The final top-1 test accuracy of ByzSGDm with various batch size under ALIE attack when coordinate-wise median (CM) and centered clipping (CC) are used as the robust aggregator

| Batch size | ByzSGDm with CM | | |
|---|---|---|---|
| | $\delta = 0$ | $\delta = 1/8$ | $\delta = 3/8$ |
| 32×8 (baseline) | **92.30%** | 86.46% | 33.11% |
| 64×8 | 91.79% (-0.51%) | 88.09% (+1.63%) | 55.66% (+22.55%) |
| 128×8 | 90.43% (-1.87%) | **89.16% (+2.70%)** | 66.38% (+33.27%) |
| 256×8 | 89.84% (-2.46%) | 88.60% (+2.14%) | 82.47% (+49.36%) |
| 512×8 | 87.27% (-5.03%) | 87.20% (+0.74%) | **83.25% (+50.14%)** |
| 1024×8 | 84.06% (-8.24%) | 83.71% (-2.75%) | 80.94% (+47.83%) |
| Batch size | ByzSGDm with CC | | |
| | $\delta = 0$ | $\delta = 1/8$ | $\delta = 3/8$ |
| 32×8 (baseline) | **92.52%** | 86.55% | 72.83% |
| 64×8 | 91.74% (-0.78%) | 88.59% (+2.04%) | 79.45% (+6.62%) |
| 128×8 | 90.63% (-1.89%) | **88.94% (+2.39%)** | 84.94% (+12.11%) |
| 256×8 | 89.40% (-3.12%) | 88.46% (+1.91%) | 87.25% (+14.42%) |
| 512×8 | 88.78% (-3.74%) | 88.29% (+1.74%) | **87.46% (+14.63%)** |
| 1024×8 | 85.50% (-7.02%) | 84.88% (-1.67%) | 83.70% (+10.87%) |

Table 3: The final top-1 test accuracy when there are 3 Byzantine workers under ALIE attack

| Method | with KR | with GM | with CM | with CC |
|---|---|---|---|---|
| ByzSGDm, batch size = $32 \times 8$ (baseline) | 38.55% | 63.11% | 33.11% | 72.83% |
| ByzSGDnm, batch size = $32 \times 8$ | 43.47% | 69.45% | 61.28% | 78.50% |
| ByzSGDm, batch size = $512 \times 8$ | 83.42% | 86.95% | 83.25% | 87.46% |
| ByzSGDnm, batch size = $512 \times 8$ | **85.12%** | **89.13%** | **86.03%** | **88.53%** |

Table 4: The final top-1 test accuracy when there are 3 Byzantine workers under FoE attack

| Method | with KR | with GM | with CM | with CC |
|---|---|---|---|---|
| ByzSGDm, batch size = $32 \times 8$ (baseline) | 10.00% | 78.36% | 83.97% | 83.60% |
| ByzSGDnm, batch size = $32 \times 8$ | 10.00% | 88.55% | 84.12% | 88.99% |
| ByzSGDm, batch size = $512 \times 8$ | 10.00% | 84.09% | 79.16% | 86.24% |
| ByzSGDnm, batch size = $512 \times 8$ | 10.00% | **89.12%** | **84.65%** | **89.32%** |

**Effectiveness of large batch size and momentum normalization.** In this paper, we propose to use (i) a relatively large batch size and (ii) the momentum normalization technique. Here, we empirically evaluate the effectiveness of these two improvements. Specifically, we will compare the performance of the following four methods: (a) ByzSGDm with batch size $32 \times 8$ (baseline), (b) ByzSGDnm with batch size $32 \times 8$, (c) ByzSGDm with batch size $512 \times 8$, and (d) ByzSGDnm with batch size $512 \times 8$. The performances of the methods are compared when there are 3 workers under ALIE attack (Baruch et al., 2019) and FoE attack (Xie et al., 2020), respectively. As presented in Table 3 and Table 4, among the four methods, ByzSGDnm with batch size $512 \times 8$ has the best top-1 test accuracy except for the case of using aggregator KR under FoE attack. All the methods fail when using KR under FoE attack mainly because the KR aggregator is not robust against FoE attack, as shown in existing works (Karimireddy et al., 2021; Xie et al., 2020). The empirical results of ByzSGDm and ByzSGDnm with more different batch size are deferred to Appendix E for space saving. In addition, we also compare the performance of ByzSGDm and ByzSGDnm under no attack (or failure) and under bit-flipping failure (Xie et al., 2019), respectively. ByzSGDnm has a comparable performance to ByzSGDm in these two cases. Please refer to Appendix E for the detailed results.

**More evaluation when NNM is used.** We also compare the empirical performance of different methods when the nearest neighbour mixing (NNM) (Allouah et al., 2023) technique is used. NNM is originally proposed to enhance the robustness of aggregators in general non-i.i.d. cases but can

Table 5: The final top-1 test accuracy when there are 3 Byzantine workers under ALIE attack and the nearest neighbour mixing (NNM) technique is used

| Method | with KR | with GM | with CM | with CC |
|---|---|---|---|---|
| ByzSGDm, batch size = $32 \times 8$ (baseline) | 58.61% | 72.58% | 71.51% | 76.48% |
| ByzSGDnm, batch size = $32 \times 8$ | 80.41% | 79.50% | 79.81% | 79.91% |
| ByzSGDm, batch size = $512 \times 8$ | 85.26% | 85.37% | 86.95% | 85.98% |
| ByzSGDnm, batch size = $512 \times 8$ | **87.68%** | **88.09%** | **87.69%** | **87.59%** |

Table 6: The wall-clock time of ByzSGDm and ByzSGDnm for 160 epochs (in second)

| Batch size | $32 \times 8$ | $64 \times 8$ | $128 \times 8$ | $256 \times 8$ | $512 \times 8$ |
|---|---|---|---|---|---|
| ByzSGDm | 2007.39s | 985.52s ($\times 2.04$ faster) | 522.27s ($\times 3.84$ faster) | 366.98s ($\times 5.47$ faster) | 314.80s ($\times 6.38$ faster) |
| ByzSGDnm | 1985.78s | 978.50s ($\times 2.03$ faster) | 515.46s ($\times 3.85$ faster) | 376.70s ($\times 5.27$ faster) | 327.62s ($\times 6.06$ faster) |

also be used in i.i.d. cases. As the results in Table 5 show, ByzSGDnm with batch size $512 \times 8$ still has the best final top-1 test accuracy under ALIE attacks when NNM is used. In addition, we find it interesting that when combined with NNM, the performance of KR and CM is improved, but the performance of GM and CC is degraded. Since NNM is originally proposed for general non-i.i.d. cases, it requires further study to understand this behavior of NNM in i.i.d. cases. However, since we mainly focus on the effect of batch size, it is beyond the scope of this work.

**The bonus of training acceleration.** Existing works (Goyal et al., 2017; Hoffer et al., 2017; Keskar et al., 2017; You et al., 2020; Zhao et al., 2020; 2023) have shown that increasing batch size can accelerate the training process by reducing the communication cost and utilizing the computing power of GPUs more effectively. We present the wall-clock time for 160 epochs when using CC as the robust aggregator under no attack in Table 6. Please note that whether there are attacks or not has almost no effect on the computation cost of non-Byzantine workers. For both ByzSGDm and ByzSGDnm, the running time decreases as the batch size increases. It verifies that increasing batch size has the bonus of training acceleration. In addition, ByzSGDnm has a comparable running time to ByzSGDm, which shows that the computation cost of the momentum normalization is negligible.

**Comparison with Byz-VR-MARINA.** Byz-VR-MARINA (Gorbunov et al., 2023) is originally proposed for non-i.i.d. cases, but can also be used in i.i.d. cases. Therefore, we also empirically compare ByzSGDnm with Byz-VR-MARINA. Empirical results show that ByzSGDnm significantly outperforms Byz-VR-MARINA in i.i.d. cases. Detailed results are deferred to Appendix D.

Although we mainly study the effect of batch size in BRDL for i.i.d. cases in this work, we also provide some empirical results under non-i.i.d. settings in Appendix E.2. The empirical results show that ByzSGDnm still outperforms existing methods in the non-i.i.d. setting. However, further work is required to detailedly discover the effect of batch size and the behavior of ByzSGDnm in non-i.i.d. cases, which we will study in the future.

## 6 CONCLUSION

In this paper, we theoretically show that when the total number of gradient computation is fixed, the optimal batch size corresponding to the tightest theoretical upper bound in BRDL increases with the fraction of Byzantine workers. The theoretical results indicate that a relatively large batch size is preferred when there are Byzantine attacks. Furthermore, we propose a novel method called ByzSGDnm and prove the convergence of ByzSGDnm. Empirical results show that when under Byzantine attacks, setting a relatively large batch size can significantly increase the model accuracy compared to the case of small batch size, which is consistent with our theoretical results. Moreover, ByzSGDnm can achieve higher model accuracy than existing BRDL methods when under attack. In addition, increasing batch size has the bonus of training acceleration, which is verified by the empirical results.

## REPRODUCIBILITY STATEMENT

For the theoretical results of our work, the assumptions are presented in Section 2 and the detailed proofs are deferred to Appendix B. For the empirical results, the experimental platform and the hyper-parameter settings are described in Section 5. The core code for our experiments can be found in the supplementary material.

## ACKNOWLEDGMENTS

This work is supported by National Key R&D Program of China (No. 2020YFA0713900), NSFC Project (No. 12326615, No. 62192783), and Fundamental Research Funds for the Central Universities (No. 020214380108).

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

# A BYZANTINE-ROBUST SGDM

The detailed algorithm of Byzantine-robust SGDm (ByzSGDm) on the server and workers are presented in Algorithm 2 and Algorithm 3, respectively.

---

**Algorithm 2** ByzSGDm (Server)

---

**Input:** worker number $m$, iteration number $T$, learning rates $\{\eta_t\}_{t=0}^{T-1}$, robust aggregator $\mathbf{Agg}(\cdot)$;
**Initialization:** model parameter $\mathbf{w}_0$;
Broadcast $\mathbf{w}_0$ to all workers;
**for** $t = 0$ **to** $T - 1$ **do**
    Receive $\{\mathbf{u}_t^{(k)}\}_{k=1}^m$ from all workers, and compute $\mathbf{u}_t = \mathbf{Agg}(\mathbf{u}_t^{(1)}, \ldots, \mathbf{u}_t^{(m)})$;
    Update model parameter $\mathbf{w}_{t+1} = \mathbf{w}_t - \eta_t \mathbf{u}_t$;
    Broadcast $\mathbf{w}_{t+1}$ to all workers;
**end for**
Output model parameter $\mathbf{w}_T$.

---

**Algorithm 3** ByzSGDm (Worker_$k$)

---

**Input:** iteration number $T$, batch size $B$, momentum hyper-parameter $\beta \in [0, 1)$;
Receive initial model parameter $\mathbf{w}_0$ from the server;
**for** $t = 0$ **to** $T - 1$ **do**
    Independently draw $B$ samples $\xi_t^{(k,1)}, \ldots, \xi_t^{(k,B)}$ from distribution $\mathcal{D}$;
    Compute $\mathbf{g}_t^{(k)} = \frac{1}{B} \sum_{b=1}^B \nabla f(\mathbf{w}_t, \xi_t^{(k,b)})$;
    Update local momentum $\mathbf{u}_t^{(k)} = \begin{cases} \mathbf{g}_0^{(k)}, & t = 0; \\ \beta \mathbf{u}_{t-1}^{(k)} + (1 - \beta)\mathbf{g}_t^{(k)}, & t > 0; \end{cases}$
    Send $\mathbf{u}_t^{(k)}$ to the server (Byzantine workers may send arbitrary values at this step);
    Receive the latest model parameter $\mathbf{w}_{t+1}$ from the server;
**end for**

---

# B PROOF DETAILS

## B.1 PROOF OF THEOREM 1

*Proof.* It has been proved in (Karimireddy et al., 2021) that for ByzSGDm, we have

$$\frac{1}{T} \sum_{t=0}^{T-1} \mathbb{E}\|\nabla F(\mathbf{w}_t)\|^2 \leq 16\sqrt{\frac{\tilde{\sigma}^2(1 + c\delta m)}{Tm}\left(10LF_0 + 3c\delta\tilde{\sigma}^2\right)} + \frac{32LF_0}{T} + \frac{20\tilde{\sigma}^2(1 + c\delta m)}{Tm}, \quad (6)$$

where $\tilde{\sigma}^2$ is the variance of stochastic gradients. In the setting of this work, we have $\tilde{\sigma}^2 = \sigma^2/B$ where $B$ is the batch size on each worker. Therefore, we have

$$\frac{1}{T} \sum_{t=0}^{T-1} \mathbb{E}\|\nabla F(\mathbf{w}_t)\|^2 \leq 16\sqrt{\frac{\sigma^2(1 + c\delta m)}{TBm}\left(10LF_0 + \frac{3c\delta\sigma^2}{B}\right)} + \frac{32LF_0}{T} + \frac{20\sigma^2(1 + c\delta m)}{TBm}. \quad (7)$$

Moreover, since for all $x, y > 0$,

$$\frac{1}{2}\sqrt{x} + \frac{1}{2}\sqrt{y} \leq \sqrt{x + y} = \sqrt{(\sqrt{x})^2 + (\sqrt{y})^2} \leq \sqrt{x} + \sqrt{y}, \quad (8)$$

we have that

$$\frac{1}{2} \leq \frac{\sqrt{x + y}}{\sqrt{x} + \sqrt{y}} \leq 1. \quad (9)$$

Therefore, we can replace the term $\sqrt{10LF_0 + \frac{3c\delta\sigma^2}{B}}$ with $\left(\sqrt{10LF_0} + \sqrt{\frac{3c\delta\sigma^2}{B}}\right)$ without changing the convergence order. Consequently, we have

$$\frac{1}{T}\sum_{t=0}^{T-1}\mathbb{E}\|\nabla F(\mathbf{w}_t)\|^2 \le 16\sqrt{\frac{\sigma^2(1+c\delta m)}{TBm}}\left(\sqrt{10LF_0} + \sqrt{\frac{3c\delta\sigma^2}{B}}\right) + \frac{32LF_0}{T} + \frac{20\sigma^2(1+c\delta m)}{TBm}. \tag{10}$$

$\square$

### B.2 Proof of Proposition 1

*Proof.* Since

$$\mathcal{U}(B) = 16\sqrt{\frac{\sigma^2(1+c\delta m)(1-\delta)}{\mathcal{C}}}\left(\sqrt{10LF_0} + \sqrt{\frac{3c\delta\sigma^2}{B}}\right) + \frac{32LF_0Bm(1-\delta)}{\mathcal{C}}$$
$$+ \frac{20\sigma^2(1+c\delta m)(1-\delta)}{\mathcal{C}}, \tag{11}$$

it is not hard to find that $\mathcal{U}(B)$ is continuous and differentiable on $(0, +\infty)$. Then we analyze the convexity of $\mathcal{U}(B)$ by showing that it has a positive second-order derivative. For simplicity, we define the constants $A_1$, $A_2$ and $A_3$ as follows:

$$A_1 = 16\sqrt{\frac{10LF_0(1+c\delta m)(1-\delta)\sigma^2}{\mathcal{C}}} + \frac{20(1+c\delta m)(1-\delta)\sigma^2}{\mathcal{C}}, \tag{12}$$

$$A_2 = 16\sqrt{\frac{3c\delta(1+c\delta m)(1-\delta)\sigma^4}{\mathcal{C}}}, \tag{13}$$

$$A_3 = \frac{32LF_0m(1-\delta)}{\mathcal{C}}. \tag{14}$$

According to the definition of $\mathcal{U}(B)$, we have

$$\mathcal{U}(B) = A_1 + A_2 B^{-\frac{1}{2}} + A_3 B, \qquad B \in (0, +\infty). \tag{15}$$

Therefore,

$$\mathcal{U}'(B) = -\frac{1}{2}A_2 B^{-\frac{3}{2}} + A_3, \qquad B \in (0, +\infty). \tag{16}$$

Thus,

$$\mathcal{U}''(B) = \frac{3}{4}A_2 B^{-\frac{5}{2}} > 0, \qquad B \in (0, +\infty). \tag{17}$$

which implies that $\mathcal{U}(B)$ is strictly convex. According to (16), the equation $\mathcal{U}'(B) = 0$ has the only solution

$$B^* = \left(\frac{A_2}{2A_3}\right)^{\frac{2}{3}} = \left(\frac{16\sqrt{\frac{3c\delta(1+c\delta m)(1-\delta)\sigma^4}{\mathcal{C}}}}{2 \cdot \frac{32LF_0m(1-\delta)}{\mathcal{C}}}\right)^{\frac{2}{3}} = \left(\frac{\sqrt{3c\delta(1+c\delta m)\sigma^4\mathcal{C}}}{4LF_0m\sqrt{1-\delta}}\right)^{\frac{2}{3}} \tag{18}$$

$$= \left(\frac{3}{16L^2(F_0)^2m}\right)^{\frac{1}{3}}\left(\frac{c\delta(1+c\delta m)}{m(1-\delta)}\right)^{\frac{1}{3}}\sigma^{\frac{4}{3}}\mathcal{C}^{\frac{1}{3}}. \tag{19}$$

Thus, $B^*$ is the only global minimizer of $\mathcal{U}(B)$. The minimum value $\mathcal{U}(B^*)$ is:

$$\mathcal{U}(B^*) = \mathcal{U}\left(\left(\frac{A_2}{2A_3}\right)^{\frac{2}{3}}\right) \tag{20}$$

$$= A_1 + 3\left(\frac{(A_2)^2 A_3}{4}\right)^{\frac{1}{3}} \tag{21}$$

$$= \frac{16\sqrt{10LF_0(1+c\delta m)(1-\delta)}\sigma}{\mathcal{C}^{\frac{1}{2}}} + \frac{20(1+c\delta m)(1-\delta)\sigma^2}{\mathcal{C}}$$

$$+ 3\left(\frac{\left(16\sqrt{\frac{3c\delta(1+c\delta m)(1-\delta)\sigma^4}{\mathcal{C}}}\right)^2 \left(\frac{32LF_0m(1-\delta)}{\mathcal{C}}\right)}{4}\right)^{\frac{1}{3}} \tag{22}$$

$$= \frac{16\sqrt{10LF_0(1+c\delta m)(1-\delta)}\sigma}{\mathcal{C}^{\frac{1}{2}}} + \frac{20(1+c\delta m)(1-\delta)\sigma^2}{\mathcal{C}}$$

$$+ \frac{24\left[12c\delta(1+c\delta m)(1-\delta)^2 LF_0m\right]^{\frac{1}{3}}\sigma^{\frac{4}{3}}}{\mathcal{C}^{\frac{2}{3}}}. \tag{23}$$

$\square$

### B.3 PROOF OF THEOREM 2

Firstly, we present Lemma 1 below, which quantitatively shows that the variance of stochastic gradients can be reduced by increasing the batch size $B$.

**Lemma 1** (Mini-batch variance reduction). *Under Assumption 1, we have that $\forall k \in \mathcal{G}$,*

$$\mathbb{E}\left\|\mathbf{g}_t^{(k)} - \nabla F(\mathbf{w}_t)\right\|^2 \leq \frac{\sigma^2}{B}, \quad \forall t \geq 0. \tag{24}$$

*Proof.* By the definition of $\mathbf{g}_t^{(k)}$,

$$\mathbf{g}_t^{(k)} = \frac{1}{B}\sum_{b=1}^{B} \nabla f(\mathbf{w}_t, \xi_t^{(k,b)}). \tag{25}$$

Since for all $k \in \mathcal{G}$, $\mathbb{E}[\nabla f(\mathbf{w}_t, \xi_t^{(k,b)})] = \nabla F(\mathbf{w}_t)$ and $\nabla f(\mathbf{w}_t, \xi_t^{(k,b)})$ is independent to each other with bounded variance $\sigma^2$, we have that

$$\mathbb{E}\left\|\mathbf{g}_t^{(k)} - \nabla F(\mathbf{w}_t)\right\|^2 \leq \frac{\sigma^2}{B}. \tag{26}$$

Moreover, by using Cauchy-Schwarz inequality, it is obtained that

$$\mathbb{E}\left\|\mathbf{g}_t^{(k)} - \nabla F(\mathbf{w}_t)\right\| \leq \sqrt{\mathbb{E}\left\|\mathbf{g}_t^{(k)} - \nabla F(\mathbf{w}_t)\right\|^2} \leq \frac{\sigma}{\sqrt{B}}. \tag{27}$$

$\square$

We define

$$\bar{\mathbf{u}}_t = \frac{1}{|\mathcal{G}|}\sum_{k\in\mathcal{G}} \mathbf{u}_t^{(k)}, \tag{28}$$

which represents the exact averaging local momentum of all non-faulty workers $k \in \mathcal{G}$. Then we provide an upper bound for the aggregation error of a $(\delta_{\max}, c)$-robust aggregator in Lemma 2. The notation of $\bar{\mathbf{u}}_t$ is only used for theoretical analysis, which does not appear in the algorithm.

The proof of Lemma 2 is inspired by the existing work (Karimireddy et al., 2021). Moreover, we have improved the analysis details, and the upper bound of aggregation error in Lemma 2 is tighter than that in (Karimireddy et al., 2021).

**Lemma 2** (Aggregation bias). *Let $\alpha = 1 - \beta$. Under Assumption 1, when $\boldsymbol{Agg}(\cdot)$ is $(\delta_{\max}, c)$-robust and $\delta \leq \delta_{\max}$, we have that*

$$\mathbb{E}\|\mathbf{u}_t - \bar{\mathbf{u}}_t\|^2 \leq \frac{2c\delta\sigma^2}{B}[\alpha + (1-\alpha)^{2t}], \quad \forall t \geq 0. \tag{29}$$

*Proof.* Let $\alpha = 1 - \beta$. For all $t \geq 0$ and fixed $k, k' \in \mathcal{G}$, since $\mathbf{g}_t^{(k)}$ is independent to $\mathbf{g}_t^{(k')}$ and $\mathbf{u}_{t-1}^{(k)}$,

$$\mathbb{E}\left\|\mathbf{u}_t^{(k)} - \mathbf{u}_t^{(k')}\right\|^2 = \mathbb{E}\left\|(1-\alpha)[\mathbf{u}_{t-1}^{(k)} - \mathbf{u}_{t-1}^{(k')}] + \alpha[\mathbf{g}_t^{(k)} - \mathbf{g}_t^{(k')}]\right\|^2 \tag{30}$$

$$= \mathbb{E}\left\|(1-\alpha)[\mathbf{u}_{t-1}^{(k)} - \mathbf{u}_{t-1}^{(k')}]\right\|^2 + \mathbb{E}\left\|\alpha[\mathbf{g}_t^{(k)} - \mathbf{g}_t^{(k')}]\right\|^2 \tag{31}$$

$$= (1-\alpha)^2 \mathbb{E}\left\|\mathbf{u}_{t-1}^{(k)} - \mathbf{u}_{t-1}^{(k')}\right\|^2 + 2\alpha^2 \mathbb{E}\left\|\mathbf{g}_t^{(k)} - \nabla F(\mathbf{w}_t)\right\|^2 \tag{32}$$

$$\leq (1-\alpha)^2 \mathbb{E}\left\|\mathbf{u}_{t-1}^{(k)} - \mathbf{u}_{t-1}^{(k')}\right\|^2 + \frac{2\alpha^2\sigma^2}{B}. \tag{33}$$

Recursively using the inequality above, we have that

$$\mathbb{E}\left\|\mathbf{u}_t^{(k)} - \mathbf{u}_t^{(k')}\right\|^2 \leq (1-\alpha)^{2t}\mathbb{E}\left\|\mathbf{u}_0^{(k)} - \mathbf{u}_0^{(k')}\right\|^2 + \frac{2\alpha^2\sigma^2}{B}[1 + (1-\alpha)^2 + \ldots + (1-\alpha)^{2(t-1)}] \tag{34}$$

$$= (1-\alpha)^{2t}\mathbb{E}\left\|\mathbf{g}_0^{(k)} - \mathbf{g}_0^{(k')}\right\|^2 + \frac{2\alpha^2\sigma^2}{B} \cdot \frac{1 - (1-\alpha)^{2t}}{1 - (1-\alpha)^2} \tag{35}$$

$$\leq (1-\alpha)^{2t} \cdot \frac{2\sigma^2}{B} + \frac{2\alpha^2\sigma^2}{B} \cdot \frac{1}{\alpha(2-\alpha)} \tag{36}$$

$$\leq (1-\alpha)^{2t} \cdot \frac{2\sigma^2}{B} + \frac{2\alpha\sigma^2}{B} \tag{37}$$

$$= \frac{2\sigma^2}{B}[\alpha + (1-\alpha)^{2t}]. \tag{38}$$

According to the definition of $(\delta_{\max}, c)$-robust aggregator,

$$\mathbb{E}\left\|\mathbf{Agg}(\mathbf{u}_t^{(1)}, \ldots, \mathbf{u}_t^{(m)}) - \frac{1}{|\mathcal{G}|}\sum_{k\in\mathcal{G}}\mathbf{u}_t^{(k)}\right\|^2 \leq \frac{2c\delta\sigma^2}{B}[\alpha + (1-\alpha)^{2t}]. \tag{39}$$

Namely,

$$\mathbb{E}\|\mathbf{u}_t - \bar{\mathbf{u}}_t\|^2 \leq \frac{2c\delta\sigma^2}{B}[\alpha + (1-\alpha)^{2t}]. \tag{40}$$

By using Cauchy-Schwarz inequality, it is obtained that

$$\mathbb{E}\|\mathbf{u}_t - \bar{\mathbf{u}}_t\| \leq \sqrt{\mathbb{E}\|\mathbf{u}_t - \bar{\mathbf{u}}_t\|^2} \leq \frac{\sqrt{2c\delta[\alpha + (1-\alpha)^{2t}]}\sigma}{\sqrt{B}}. \tag{41}$$

$\square$

Based on Lemma 2, we can further obtain Lemma 3, which provides an upper bound for the difference between the aggregated momentum $\mathbf{u}_t$ and the global gradient $\nabla F(\mathbf{w}_t)$.

**Lemma 3.** *Under Assumptions 1 and 3, when $\mathbf{Agg}(\cdot)$ is $(\delta_{\max}, c)$-robust, $\delta \leq \delta_{\max}$ and $\eta_t = \eta$, we have the following result for ByzSGDnm:*

$$\mathbb{E}\|\mathbf{u}_t - \nabla F(\mathbf{w}_t)\| \leq \frac{\eta L}{\alpha} + \frac{\sqrt{2cm\delta(1-\delta)} + 1}{\sqrt{Bm(1-\delta)}}\left[(1-\alpha)^t + \sqrt{\alpha}\right]\sigma, \quad \forall t \geq 0. \tag{42}$$

*Proof.* $\forall t \geq 0$, we have that

$$\bar{\mathbf{u}}_t - \nabla F(\mathbf{w}_t) = \left(\frac{1}{|\mathcal{G}|}\sum_{k\in\mathcal{G}}\mathbf{u}_t^{(k)}\right) - \nabla F(\mathbf{w}_t) \tag{43}$$

$$= \frac{1}{|\mathcal{G}|}\sum_{k\in\mathcal{G}}\left[\beta\mathbf{u}_{t-1}^{(k)} + (1-\beta)\mathbf{g}_t^{(k)}\right] - \nabla F(\mathbf{w}_t) \tag{44}$$

$$= \beta \bar{\mathbf{u}}_{t-1} + \left( \frac{1-\beta}{|\mathcal{G}|} \sum_{k \in \mathcal{G}} \mathbf{g}_t^{(k)} \right) - \nabla F(\mathbf{w}_t) \tag{45}$$

$$= \beta [\bar{\mathbf{u}}_{t-1} - \nabla F(\mathbf{w}_t)] + \frac{1-\beta}{|\mathcal{G}|} \sum_{k \in \mathcal{G}} \left[ \mathbf{g}_t^{(k)} - \nabla F(\mathbf{w}_t) \right] \tag{46}$$

$$= \beta [\bar{\mathbf{u}}_{t-1} - \nabla F(\mathbf{w}_{t-1})] + \beta [\nabla F(\mathbf{w}_t) - \nabla F(\mathbf{w}_{t-1})] + \frac{1-\beta}{|\mathcal{G}|} \sum_{k \in \mathcal{G}} \left[ \mathbf{g}_t^{(k)} - \nabla F(\mathbf{w}_t) \right]. \tag{47}$$

Recursively using the equation above and substituting $\beta$ with $1 - \alpha$, we have that

$$\bar{\mathbf{u}}_t - \nabla F(\mathbf{w}_t) = (1-\alpha)^t [\bar{\mathbf{u}}_0 - \nabla F(\mathbf{w}_0)] + \sum_{t'=1}^{t} (1-\alpha)^{t-t'+1} [\nabla F(\mathbf{w}_{t'}) - \nabla F(\mathbf{w}_{t'-1})]$$

$$+ \sum_{t'=1}^{t} (1-\alpha)^{t-t'} \left\{ \frac{\alpha}{|\mathcal{G}|} \sum_{k \in \mathcal{G}} \left[ \mathbf{g}_{t'}^{(k)} - \nabla F(\mathbf{w}_{t'}) \right] \right\}. \tag{48}$$

Noticing that $\mathbf{u}_0^{(k)} = \mathbf{g}_0^{(k)}$, we have that $\bar{\mathbf{u}}_0 - \nabla F(\mathbf{w}_0) = \frac{1}{|\mathcal{G}|} \sum_{k \in \mathcal{G}} [\mathbf{g}_0^{(k)} - \nabla F(\mathbf{w}_0)]$. Therefore,

$$\bar{\mathbf{u}}_t - \nabla F(\mathbf{w}_t) = \sum_{t'=1}^{t} (1-\alpha)^{t-t'+1} [\nabla F(\mathbf{w}_{t'}) - \nabla F(\mathbf{w}_{t'-1})]$$

$$+ \frac{1}{|\mathcal{G}|} \sum_{k \in \mathcal{G}} \left\{ (1-\alpha)^t [\mathbf{g}_0^{(k)} - \nabla F(\mathbf{w}_0)] + \alpha \sum_{t'=1}^{t} (1-\alpha)^{t-t'} [\mathbf{g}_{t'}^{(k)} - \nabla F(\mathbf{w}_{t'})] \right\}. \tag{49}$$

According to Assumption 3,

$$\mathbb{E} \| \nabla F(\mathbf{w}_{t'}) - \nabla F(\mathbf{w}_{t'-1}) \| \le L \cdot \mathbb{E} \| \mathbf{w}_{t'} - \mathbf{w}_{t'-1} \| = L \cdot \mathbb{E} \left\| -\eta \frac{\mathbf{u}_{t'-1}}{\| \mathbf{u}_{t'-1} \|} \right\| = \eta L. \tag{50}$$

Since $\mathbf{g}_t^{(k)}$'s are independent to each other, by using Lemma 1 and $|\mathcal{G}| \ge (1-\delta)m$, we have that

$$\mathbb{E} \left\| \frac{1}{|\mathcal{G}|} \sum_{k \in \mathcal{G}} \left\{ (1-\alpha)^t [\mathbf{g}_0^{(k)} - \nabla F(\mathbf{w}_0)] + \alpha \sum_{t'=1}^{t} (1-\alpha)^{t-t'} [\mathbf{g}_{t'}^{(k)} - \nabla F(\mathbf{w}_{t'})] \right\} \right\|$$

$$\le \left( \mathbb{E} \left\| \frac{1}{|\mathcal{G}|} \sum_{k \in \mathcal{G}} \left\{ (1-\alpha)^t [\mathbf{g}_0^{(k)} - \nabla F(\mathbf{w}_0)] + \alpha \sum_{t'=1}^{t} (1-\alpha)^{t-t'} [\mathbf{g}_{t'}^{(k)} - \nabla F(\mathbf{w}_{t'})] \right\} \right\|^2 \right)^{\frac{1}{2}} \tag{51}$$

$$= \frac{1}{|\mathcal{G}|} \left\{ \sum_{k \in \mathcal{G}} \left[ (1-\alpha)^{2t} \mathbb{E} \left\| \mathbf{g}_0^{(k)} - \nabla F(\mathbf{w}_0) \right\|^2 + \alpha^2 \sum_{t'=1}^{t} (1-\alpha)^{2t-2t'} \mathbb{E} \left\| \mathbf{g}_{t'}^{(k)} - \nabla F(\mathbf{w}_{t'}) \right\|^2 \right] \right\}^{\frac{1}{2}} \tag{52}$$

$$\le \frac{1}{|\mathcal{G}|} \left\{ \frac{|\mathcal{G}|\sigma^2}{B} \left[ (1-\alpha)^{2t} + \alpha^2 \sum_{t'=1}^{t} (1-\alpha)^{2t-2t'} \right] \right\}^{\frac{1}{2}} \tag{53}$$

$$= \frac{\sigma}{\sqrt{B|\mathcal{G}|}} \sqrt{(1-\alpha)^{2t} + \alpha^2 \frac{1-(1-\alpha)^{2t}}{1-(1-\alpha)^2}} \tag{54}$$

$$\le \frac{\sigma}{\sqrt{B|\mathcal{G}|}} \left[ (1-\alpha)^t + \alpha \sqrt{\frac{1-(1-\alpha)^{2t}}{1-(1-\alpha)^2}} \right] \tag{55}$$

$$\leq \frac{\sigma}{\sqrt{B|\mathcal{G}|}} \left[ (1-\alpha)^t + \alpha\sqrt{\frac{1}{\alpha(2-\alpha)}} \right] \tag{56}$$

$$= \frac{\sigma}{\sqrt{B|\mathcal{G}|}} \left[ (1-\alpha)^t + \sqrt{\frac{\alpha}{2-\alpha}} \right] \tag{57}$$

$$\leq \frac{\sigma}{\sqrt{Bm(1-\delta)}} \left[ (1-\alpha)^t + \sqrt{\alpha} \right]. \tag{58}$$

Consequently,

$$\mathbb{E}\|\bar{\mathbf{u}}_t - \nabla F(\mathbf{w}_t)\| \leq \eta L \sum_{t'=1}^{t} (1-\alpha)^{t-t'+1} + \frac{\sigma}{\sqrt{Bm(1-\delta)}} \left[ (1-\alpha)^t + \sqrt{\alpha} \right] \tag{59}$$

$$\leq \frac{\eta L}{\alpha} + \frac{\sigma}{\sqrt{Bm(1-\delta)}} \left[ (1-\alpha)^t + \sqrt{\alpha} \right]. \tag{60}$$

Since $\mathbb{E}\|\mathbf{u}_t - \bar{\mathbf{u}}_t\| \leq \sqrt{\mathbb{E}\|\mathbf{u}_t - \bar{\mathbf{u}}_t\|^2}$, according to Lemma 2, we have that

$$\mathbb{E}\|\mathbf{u}_t - \nabla F(\mathbf{w}_t)\| \leq \mathbb{E}\|\bar{\mathbf{u}}_t - \nabla F(\mathbf{w}_t)\| + \mathbb{E}\|\mathbf{u}_t - \bar{\mathbf{u}}_t\| \tag{61}$$

$$\leq \frac{\eta L}{\alpha} + \frac{\sigma}{\sqrt{Bm(1-\delta)}} \left[ (1-\alpha)^t + \sqrt{\alpha} \right] + \frac{\sqrt{2c\delta[\alpha + (1-\alpha)^{2t}]}\sigma}{\sqrt{B}} \tag{62}$$

$$\leq \frac{\eta L}{\alpha} + \frac{\sigma}{\sqrt{Bm(1-\delta)}} \left[ (1-\alpha)^t + \sqrt{\alpha} \right] + \frac{\sqrt{2c\delta}[\sqrt{\alpha} + (1-\alpha)^t]\sigma}{\sqrt{B}} \tag{63}$$

$$= \frac{\eta L}{\alpha} + \frac{\sqrt{2cm\delta(1-\delta)} + 1}{\sqrt{Bm(1-\delta)}} \left[ (1-\alpha)^t + \sqrt{\alpha} \right] \sigma. \tag{64}$$

$$\square$$

Then we present the descent lemma for SGD with normalized momentum. The proof of Lemma 4 is inspired by the existing work (Cutkosky & Mehta, 2020), but the result in Lemma 4 is more general than that in (Cutkosky & Mehta, 2020).

**Lemma 4** (Descent lemma). *Under Assumptions 1 and 3, for any constant $\gamma \in (0, 1)$, we have the following result for ByzSGDnm:*

$$F(\mathbf{w}_{t+1}) \leq F(\mathbf{w}_t) - \eta_t \frac{1-\gamma}{1+\gamma} \|\nabla F(\mathbf{w}_t)\| + \eta_t \frac{2}{\gamma(1+\gamma)} \|\mathbf{u}_t - \nabla F(\mathbf{w}_t)\| + \frac{(\eta_t)^2 L}{2}. \tag{65}$$

*Proof.* According to Assumption 3, we have that

$$F(\mathbf{w}_{t+1}) = F(\mathbf{w}_t - \eta_t \frac{\mathbf{u}_t}{\|\mathbf{u}_t\|}) \tag{66}$$

$$\leq F(\mathbf{w}_t) - \nabla F(\mathbf{w}_t)^T \left( \eta_t \frac{\mathbf{u}_t}{\|\mathbf{u}_t\|} \right) + \frac{L}{2} \left\| \eta_t \frac{\mathbf{u}_t}{\|\mathbf{u}_t\|} \right\|^2 \tag{67}$$

$$= F(\mathbf{w}_t) - \eta_t \frac{\nabla F(\mathbf{w}_t)^T \mathbf{u}_t}{\|\mathbf{u}_t\|} + \frac{(\eta_t)^2 L}{2}. \tag{68}$$

Let $\gamma \in (0, 1)$ be an arbitrary constant. Then we consider the following two cases:

(i) When $\|\mathbf{u}_t - \nabla F(\mathbf{w}_t)\| \leq \gamma\|\nabla F(\mathbf{w}_t)\|$, we have

$$-\eta_t \frac{\nabla F(\mathbf{w}_t)^T \mathbf{u}_t}{\|\mathbf{u}_t\|} = -\eta_t \frac{\nabla F(\mathbf{w}_t)^T [\nabla F(\mathbf{w}_t) + (\mathbf{u}_t - \nabla F(\mathbf{w}_t))]}{\|\nabla F(\mathbf{w}_t) + (\mathbf{u}_t - \nabla F(\mathbf{w}_t))\|} \tag{69}$$

$$\leq -\eta_t \frac{\|\nabla F(\mathbf{w}_t)\|^2 - \|\nabla F(\mathbf{w}_t)\| \cdot \|\mathbf{u}_t - \nabla F(\mathbf{w}_t)\|}{\|\nabla F(\mathbf{w}_t)\| + \|\mathbf{u}_t - \nabla F(\mathbf{w}_t)\|} \tag{70}$$

$$\leq - \eta_t \frac{\|\nabla F(\mathbf{w}_t)\|^2 - \gamma \|\nabla F(\mathbf{w}_t)\|^2}{(1+\gamma)\|\nabla F(\mathbf{w}_t)\|} \tag{71}$$

$$= - \eta_t \frac{1-\gamma}{1+\gamma} \|\nabla F(\mathbf{w}_t)\| \tag{72}$$

$$\leq - \eta_t \frac{1-\gamma}{1+\gamma} \|\nabla F(\mathbf{w}_t)\| + \eta_t \frac{2}{\gamma(1+\gamma)} \|\mathbf{u}_t - \nabla F(\mathbf{w}_t)\|. \tag{73}$$

(ii) When $\|\mathbf{u}_t - \nabla F(\mathbf{w}_t)\| > \gamma \|\nabla F(\mathbf{w}_t)\|$, we have

$$-\eta_t \frac{\nabla F(\mathbf{w}_t)^T \mathbf{u}_t}{\|\mathbf{u}_t\|} \leq \eta_t \|\nabla F(\mathbf{w}_t)\| \tag{74}$$

$$= - \eta_t \frac{1-\gamma}{1+\gamma} \|\nabla F(\mathbf{w}_t)\| + \eta_t \frac{2}{1+\gamma} \|\nabla F(\mathbf{w}_t)\| \tag{75}$$

$$\leq - \eta_t \frac{1-\gamma}{1+\gamma} \|\nabla F(\mathbf{w}_t)\| + \eta_t \frac{2}{\gamma(1+\gamma)} \|\mathbf{u}_t - \nabla F(\mathbf{w}_t)\|. \tag{76}$$

In summary, we always have

$$- \eta_t \frac{\nabla F(\mathbf{w}_t)^T \mathbf{u}_t}{\|\mathbf{u}_t\|} \leq -\eta_t \frac{1-\gamma}{1+\gamma} \|\nabla F(\mathbf{w}_t)\| + \eta_t \frac{2}{\gamma(1+\gamma)} \|\mathbf{u}_t - \nabla F(\mathbf{w}_t)\|. \tag{77}$$

Therefore,

$$F(\mathbf{w}_{t+1}) \leq F(\mathbf{w}_t) - \eta_t \frac{1-\gamma}{1+\gamma} \|\nabla F(\mathbf{w}_t)\| + \eta_t \frac{2}{\gamma(1+\gamma)} \|\mathbf{u}_t - \nabla F(\mathbf{w}_t)\| + \frac{(\eta_t)^2 L}{2}. \tag{78}$$

$\square$

Finally, we can obtain Theorem 2 by recursively using Lemma 4, taking expectation on both sides and using Lemma 3. The proof details are presented below.

*Proof.* Recursively using Lemma 4 from $t=0$ to $T-1$ and letting $\eta_t = \eta$, we have that

$$F(\mathbf{w}_T) \leq F(\mathbf{w}_0) - \frac{(1-\gamma)\eta}{1+\gamma} \sum_{t=0}^{T-1} \|\nabla F(\mathbf{w}_t)\| + \frac{2\eta}{\gamma(1+\gamma)} \sum_{t=0}^{T-1} \|\mathbf{u}_t - \nabla F(\mathbf{w}_t)\| + \frac{T\eta^2 L}{2}. \tag{79}$$

Therefore,

$$\frac{1}{T} \sum_{t=0}^{T-1} \|\nabla F(\mathbf{w}_t)\| \leq \frac{(1+\gamma)[F(\mathbf{w}_0) - F(\mathbf{w}_T)]}{(1-\gamma)\eta T} + \frac{2}{T\gamma(1-\gamma)} \sum_{t=0}^{T-1} \|\mathbf{u}_t - \nabla F(\mathbf{w}_t)\| + \frac{(1+\gamma)\eta L}{2(1-\gamma)}. \tag{80}$$

According to Assumption 2, we have that $F(\mathbf{w}_T) \geq F^*$. Furthermore, by taking expectation on both sides, letting $\gamma = \frac{1}{3}$ and using Lemma 3, it is obtained that

$$\frac{1}{T} \sum_{t=0}^{T-1} \mathbb{E}\|\nabla F(\mathbf{w}_t)\| \leq \frac{2[F(\mathbf{w}_0) - F^*]}{\eta T} + \frac{9}{T} \sum_{t=0}^{T-1} \mathbb{E}\|\mathbf{u}_t - \nabla F(\mathbf{w}_t)\| + \eta L \tag{81}$$

$$\leq \frac{2[F(\mathbf{w}_0) - F^*]}{\eta T} + \frac{9\eta L}{\alpha} + \frac{9\sqrt{2cm\delta(1-\delta)} + 9}{\sqrt{Bm(1-\delta)}} \left( \frac{1}{\alpha T} + \sqrt{\alpha} \right) \sigma + \eta L \tag{82}$$

$$\leq \frac{2F_0}{\eta T} + \frac{10\eta L}{\alpha} + \frac{9\sqrt{2cm\delta(1-\delta)} + 9}{\sqrt{Bm(1-\delta)}} \left( \frac{1}{\alpha T} + \sqrt{\alpha} \right) \sigma. \tag{83}$$

$\square$

## B.4 PROOF OF PROPOSITION 2

*Proof.* Learning rate $\eta$ appears only in the first two terms of the RHS of inequality (2). Thus,

$$\frac{1}{T}\sum_{t=0}^{T-1}\mathbb{E}\|\nabla F(\mathbf{w}_t)\| \leq \frac{2F_0}{\eta T} + \frac{10\eta L}{\alpha} + \frac{9\sqrt{2cm\delta(1-\delta)}+9}{\sqrt{Bm(1-\delta)}}\left(\frac{1}{\alpha T}+\sqrt{\alpha}\right)\sigma$$

$$\leq 2\sqrt{\frac{2F_0}{\eta T}\times\frac{10\eta L}{\alpha}} + \frac{9\sqrt{2cm\delta(1-\delta)}+9}{\sqrt{Bm(1-\delta)}}\left(\frac{1}{\alpha T}+\sqrt{\alpha}\right)\sigma \qquad (84)$$

$$= \sqrt{\frac{80LF_0}{\alpha T}} + \frac{9\sqrt{2cm\delta(1-\delta)}+9}{\sqrt{Bm(1-\delta)}}\left(\frac{1}{\alpha T}+\sqrt{\alpha}\right)\sigma. \qquad (85)$$

The second equation holds only and only if $\frac{2F_0}{\eta T} = \frac{10\eta L}{\alpha}$, which is equivalent to that $\eta = \sqrt{\frac{\alpha F_0}{5LT}}$.
Furthermore, since $\alpha = \min\left(\frac{\sqrt{80LF_0Bm(1-\delta)}}{\left[9\sqrt{2cm\delta(1-\delta)}+9\right]\sigma\sqrt{T}}, 1\right)$, we consider the following two cases.

(i) When $\alpha = \frac{\sqrt{80LF_0Bm(1-\delta)}}{\left[9\sqrt{2cm\delta(1-\delta)}+9\right]\sigma\sqrt{T}}$, we have that

$$\frac{1}{T}\sum_{t=0}^{T-1}\mathbb{E}\|\nabla F(\mathbf{w}_t)\| \leq 6\left[\sqrt{2cm\delta(1-\delta)}+1\right]^{\frac{1}{2}}\left(\frac{5LF_0\sigma^2}{TBm(1-\delta)}\right)^{\frac{1}{4}}$$

$$+ \frac{27\left[\sqrt{2cm\delta(1-\delta)}+1\right]^{\frac{3}{2}}}{4\sqrt{5TB^2m^2(1-\delta)^2LF_0}}\sigma^2. \qquad (86)$$

(ii) When $\alpha = 1$, it implies $\frac{\sqrt{80LF_0Bm(1-\delta)}}{\left[9\sqrt{2cm\delta(1-\delta)}+9\right]\sigma\sqrt{T}} \geq 1$. Namely, $\frac{9\sqrt{2cm\delta(1-\delta)}+9}{\sqrt{Bm(1-\delta)}}\sigma \leq \frac{\sqrt{80LF_0}}{\sqrt{T}}$. In this case,

$$\frac{1}{T}\sum_{t=0}^{T-1}\mathbb{E}\|\nabla F(\mathbf{w}_t)\| \leq \sqrt{\frac{80LF_0}{\alpha T}} + \frac{9\sqrt{2cm\delta(1-\delta)}+9}{\sqrt{Bm(1-\delta)}}\left(\frac{1}{\alpha T}+\sqrt{\alpha}\right)\sigma \qquad (87)$$

$$\leq \sqrt{\frac{80LF_0}{T}} + \sqrt{\frac{80LF_0}{T}}\left(\frac{1}{T}+1\right) \qquad (88)$$

$$\leq 12\sqrt{\frac{5LF_0}{T}}. \qquad (89)$$

In summary,

$$\frac{1}{T}\sum_{t=0}^{T-1}\mathbb{E}\|\nabla F(\mathbf{w}_t)\| \leq 6\left[\sqrt{2cm\delta(1-\delta)}+1\right]^{\frac{1}{2}}\left(\frac{5LF_0\sigma^2}{TBm(1-\delta)}\right)^{\frac{1}{4}} + 12\sqrt{\frac{5LF_0}{T}}$$

$$+ \frac{27\left[\sqrt{2cm\delta(1-\delta)}+1\right]^{\frac{3}{2}}}{4\sqrt{5TB^2m^2(1-\delta)^2LF_0}}\sigma^2. \qquad (90)$$

In addition, when $\mathcal{C} = TBm(1-\delta)$ is fixed, we have

$$\frac{1}{T}\sum_{t=0}^{T-1}\mathbb{E}\|\nabla F(\mathbf{w}_t)\| \leq 6\left[\sqrt{2cm\delta(1-\delta)}+1\right]^{\frac{1}{2}}\left(\frac{5LF_0\sigma^2}{\mathcal{C}}\right)^{\frac{1}{4}}$$

$$+ 12\sqrt{\frac{5m(1-\delta)LF_0B}{\mathcal{C}}} + \frac{27\left[\sqrt{2cm\delta(1-\delta)}+1\right]^{\frac{3}{2}}}{4\sqrt{5m(1-\delta)LF_0\mathcal{C}B}}\sigma^2. \qquad (91)$$

Moreover,

$$12\sqrt{\frac{5m(1-\delta)LF_0B}{\mathcal{C}}} + \frac{27\left[\sqrt{2cm\delta(1-\delta)}+1\right]^{\frac{3}{2}}}{4\sqrt{5m(1-\delta)LF_0\mathcal{C}B}}\sigma^2$$

$$\geq 2\sqrt{\left(12\sqrt{\frac{5m(1-\delta)LF_0B}{\mathcal{C}}}\right) \times \left(\frac{27\left[\sqrt{2cm\delta(1-\delta)}+1\right]^{\frac{3}{2}}}{4\sqrt{5m(1-\delta)LF_0\mathcal{C}B}}\sigma^2\right)} \tag{92}$$

$$= \frac{18\left[\sqrt{2cm\delta(1-\delta)}+1\right]^{\frac{3}{4}}\sigma}{\mathcal{C}^{\frac{1}{2}}}. \tag{93}$$

The equation holds if and only if

$$12\sqrt{\frac{5m(1-\delta)LF_0B}{\mathcal{C}}} = \frac{27\left[\sqrt{2cm\delta(1-\delta)}+1\right]^{\frac{3}{2}}}{4\sqrt{5m(1-\delta)LF_0\mathcal{C}B}}\sigma^2, \tag{94}$$

which is equivalent to that

$$B = \frac{9\left[\sqrt{2cm\delta(1-\delta)}+1\right]^{\frac{3}{2}}\sigma^2}{80m(1-\delta)LF_0}. \tag{95}$$

$\square$

## C    DISCUSSION ABOUT THE $(f, \kappa)$-ROBUSTNESS

The theoretical analysis in the main text is based on the definition of $(\delta_{\max}, c)$-robust aggregator (Karimireddy et al., 2021). In this section, we show that similar results can be obtained under the definition of $(f, \kappa)$-robustness (Allouah et al., 2023). In existing works 11, $f$ is defined to be the number of Byzantine workers. However, we have used the notation $f$ to denote the loss function in this paper. In order to avoid misunderstanding, we will use $m\delta$ to denote the number of Byzantine workers in the following text, where $m$ is the total number of workers and $\delta$ is the fraction of Byzantine workers. Firstly, we present the definition of $(\delta, \kappa)$-robust aggregator, which is equivalent to the $(f, \kappa)$-robustness (Allouah et al., 2023) since the worker number $m$ is deterministic in this paper.

**Definition 2** ($(\delta, \kappa)$-robustness)**.** *Let $\delta \in [0, \frac{1}{2})$ and $\kappa \geq 0$. An aggregator $\mathbf{Agg}(\cdot)$ is called a $(\delta, \kappa)$-robust aggregator if for any vectors $\mathbf{x}_1, \ldots, \mathbf{x}_m \in \mathbb{R}^d$ and any set $\mathcal{G} \subseteq \{1, \ldots, m\}$ satisfying $|\mathcal{G}| = (1-\delta)m$, we have that*

$$\|\mathbf{Agg}(\mathbf{x}_1, \ldots, \mathbf{x}_m) - \bar{\mathbf{x}}_{\mathcal{G}}\|^2 \leq \frac{\kappa}{|\mathcal{G}|}\sum_{k \in \mathcal{G}}\|\mathbf{x}_k - \bar{\mathbf{x}}_{\mathcal{G}}\|^2, \tag{96}$$

*where $\bar{\mathbf{x}}_{\mathcal{G}} = \frac{1}{|\mathcal{G}|}\sum_{k \in \mathcal{G}}\mathbf{x}_k$.*

Thus, for a $(\delta, \kappa)$-robust aggregator $\mathbf{Agg}(\cdot)$, when $\{\mathbf{x}_k\}_{k \in \mathcal{G}'}$ is a set of i.i.d. random vectors where $\mathcal{G} \subseteq \{1, \ldots, m\}$ satisfying $|\mathcal{G}| = (1-\delta)m$ and $\mathbb{E}\|\mathbf{x}_k - \mathbb{E}[\mathbf{x}_k]\|^2 = \sigma^2$ for each $k \in \mathcal{G}$, we have:

$$\mathbb{E}\|\mathbf{Agg}(\mathbf{x}_1, \ldots, \mathbf{x}_m) - \bar{\mathbf{x}}_{\mathcal{G}}\|^2 \leq \mathbb{E}\left[\frac{\kappa}{|\mathcal{G}|}\sum_{k \in \mathcal{G}}\|\mathbf{x}_k - \bar{\mathbf{x}}_{\mathcal{G}}\|^2\right] = \frac{|\mathcal{G}|-1}{|\mathcal{G}|}\kappa\sigma^2 = \left(1 - \frac{1}{(1-\delta)m}\right)\kappa\sigma^2. \tag{97}$$

Meanwhile, for a $(\delta_{\max}, c)$-robust aggregator $\mathbf{Agg}(\cdot)$, when $\{\mathbf{x}_k\}_{k \in \mathcal{G}'}$ is a set of i.i.d. random vectors where $\mathcal{G} \subseteq \{1, \ldots, m\}$ satisfying $|\mathcal{G}| = (1-\delta)m$ and $\mathbb{E}\|\mathbf{x}_k - \mathbb{E}[\mathbf{x}_k]\|^2 = \sigma^2$ for each $k \in \mathcal{G}$, we have $\mathbb{E}\|\mathbf{x}_k - \mathbf{x}_{k'}\|^2 \leq 2\sigma^2$ according to (2), and thus

$$\mathbb{E}\|\mathbf{Agg}(\mathbf{x}_1, \ldots, \mathbf{x}_m) - \bar{\mathbf{x}}_{\mathcal{G}}\|^2 \leq 2c\delta\sigma^2. \tag{98}$$

Comparing (97) and (98), we can find that in i.i.d. cases, the upper bounds of aggregation error under the two definitions are of the same order $O(\sigma^2)$. Moreover, the other factors $|\mathcal{G}|$, $\kappa$, $c$ and $\delta$ will not change during the distributed learning process. Thus, to obtain the optimal batch size that minimizes the theoretical upper bound under the $(\delta, \kappa)$-robustness, we can simply replace the factor $c\delta$ with $\tilde{c} = \frac{\kappa}{2}\left(1 - \frac{1}{(1-\delta)m}\right)$.

Recall that the optimal batch size that minimize the theoretical upper bound for ByzSGDm and ByzSGDnm under the definition of $(\delta_{\max}, c)$-robustness are

$$B^* = \left(\frac{3}{16L^2(F_0)^2 m}\right)^{\frac{1}{3}}\left(\frac{c\delta(1 + c\delta m)}{m(1-\delta)}\right)^{\frac{1}{3}}\sigma^{\frac{4}{3}}\mathcal{C}^{\frac{1}{3}}, \tag{99}$$

and

$$\tilde{B}^* = \frac{9\left[\sqrt{2cm\delta(1-\delta)} + 1\right]^{\frac{3}{2}}\sigma^2}{80m(1-\delta)LF_0} = \frac{9\left[\frac{\sqrt{2cm\delta}}{(1-\delta)^{\frac{1}{6}}} + \frac{1}{(1-\delta)^{\frac{2}{3}}}\right]^{\frac{3}{2}}\sigma^2}{80mLF_0}, \tag{100}$$

respectively. Thus, under the definition of $(\delta, \kappa)$-robustness, the optimal batch size that minimize the theoretical upper bound for ByzSGDm and ByzSGDnm are

$$\left(\frac{3}{16L^2(F_0)^2 m}\right)^{\frac{1}{3}}\left(\frac{\tilde{c}(1 + \tilde{c}m)}{m(1-\delta)}\right)^{\frac{1}{3}}\sigma^{\frac{4}{3}}\mathcal{C}^{\frac{1}{3}},$$

and

$$\frac{9\left[\frac{\sqrt{2\tilde{c}m}}{(1-\delta)^{\frac{1}{6}}} + \frac{1}{(1-\delta)^{\frac{2}{3}}}\right]^{\frac{3}{2}}\sigma^2}{80mLF_0},$$

respectively. Notice that the term $\frac{1}{1-\delta}$ is monotonically increasing w.r.t. $\delta$. Thus, in order to prove that the optimal batch size is monotonically increasing w.r.t. $\delta$, we only need to prove that $\tilde{c} = \frac{\kappa}{2}\left(1 - \frac{1}{(1-\delta)m}\right)$ is monotonically increasing w.r.t. $\delta$.

It has been shown in existing works (Allouah et al., 2023) that (i) $\kappa = 1 + \frac{\delta}{1-2\delta}$ for Krum, (ii) $\kappa = \frac{\delta}{1-2\delta}(1 + \frac{\delta}{1-2\delta})$ for coordinate-wise trimmed-mean, (iii) $\kappa = (1 + \frac{\delta}{1-2\delta})^2$ for coordinate-wise median and geometric median, and that (iv) the lower bound for $\kappa$ is $\kappa = \frac{\delta}{1-2\delta}$. Then we prove that $\tilde{c}$ is monotonically increasing w.r.t. $\delta$ for these four cases separately.

Case (i). For Krum, we have $\kappa = 1 + \frac{\delta}{1-2\delta}$ $(0 \le \delta < \frac{1}{2})$, and thus,

$$\tilde{c}(\delta) = \frac{1}{2}\left(1 + \frac{\delta}{1-2\delta}\right)\left(1 - \frac{1}{(1-\delta)m}\right) \tag{101}$$

$$= \frac{1}{2}\cdot\frac{1-\delta}{1-2\delta}\left(1 - \frac{1}{(1-\delta)m}\right) \tag{102}$$

$$= \frac{1}{2}\cdot\frac{1-\delta-\frac{1}{m}}{1-2\delta}. \tag{103}$$

Therefore, the derivative of $\tilde{c}(\delta)$ is

$$\tilde{c}'(\delta) = \frac{1}{2}\cdot\frac{-(1-2\delta) + 2(1-\delta-\frac{1}{m})}{(1-2\delta)^2} = \frac{m-2}{2m(1-2\delta)^2}. \tag{104}$$

Since the fraction of Byzantine workers is smaller than $\frac{1}{2}$, Byzantine worker can appear only when the total worker number $m > 2$. Thus, we have $m - 2 > 0$. In addition, $(1-2\delta)^2 > 0$ since $0 \le \delta < \frac{1}{2}$. Therefore, we have $\tilde{c}'(\delta) > 0$, and $\tilde{c}(\delta)$ is monotonically increasing w.r.t. $\delta$.

Case (ii). For coordinate-wise trimmed-mean, we have $\kappa = \frac{\delta}{1-2\delta}(1 + \frac{\delta}{1-2\delta})$. Thus,

$$\tilde{c}(\delta) = \frac{1}{2}\cdot\frac{\delta}{1-2\delta}\left(1 + \frac{\delta}{1-2\delta}\right)\left(1 - \frac{1}{(1-\delta)m}\right) \tag{105}$$

$$= \left[ \frac{1}{2} \left( 1 + \frac{\delta}{1 - 2\delta} \right) \left( 1 - \frac{1}{(1 - \delta)m} \right) \right] \cdot \left( \frac{\delta}{1 - 2\delta} \right). \tag{106}$$

It has been proven in case (i) that the first term $\left[ \frac{1}{2} \left( 1 + \frac{\delta}{1-2\delta} \right) \left( 1 - \frac{1}{(1-\delta)m} \right) \right]$ is monotonically increasing w.r.t. $\delta$. Since both $\delta$ and $\frac{1}{1-2\delta}$ increase as $\delta \in [0, \frac{1}{2})$ increases, the second term $\frac{\delta}{1-2\delta}$ is also monotonically increasing w.r.t. $\delta$. Moreover, the two terms are both positive. Thus, we have that $\tilde{c}(\delta)$ is monotonically increasing w.r.t. $\delta$.

Case (iii). For coordinate-wise median and geometric median, we have $\kappa = (1 + \frac{\delta}{1-2\delta})^2$. Thus,

$$\tilde{c}(\delta) = \frac{1}{2} \cdot \left( 1 + \frac{\delta}{1 - 2\delta} \right)^2 \left( 1 - \frac{1}{(1 - \delta)m} \right) \tag{107}$$

$$= \left[ \frac{1}{2} \left( 1 + \frac{\delta}{1 - 2\delta} \right) \left( 1 - \frac{1}{(1 - \delta)m} \right) \right] \cdot \left( 1 + \frac{\delta}{1 - 2\delta} \right). \tag{108}$$

It has been proven in case (i) that the first term $\left[ \frac{1}{2} \left( 1 + \frac{\delta}{1-2\delta} \right) \left( 1 - \frac{1}{(1-\delta)m} \right) \right]$ is monotonically increasing w.r.t. $\delta$. Since both $\delta$ and $\frac{1}{1-2\delta}$ increase as $\delta \in [0, \frac{1}{2})$ increases, the second term $\left( 1 + \frac{\delta}{1-2\delta} \right)$ is also monotonically increasing w.r.t. $\delta$. Moreover, the two terms are both positive. Thus, we have that $\tilde{c}(\delta)$ is monotonically increasing w.r.t. $\delta$.

Case (iv). The lower bound for $\kappa$ is $\kappa = \frac{\delta}{1-2\delta}$. Thus, the lower bound for $\tilde{c}(\delta)$ is

$$\tilde{c}(\delta) = \frac{1}{2} \cdot \frac{\delta}{1 - 2\delta} \left( 1 - \frac{1}{(1 - \delta)m} \right) \tag{109}$$

$$= \frac{1}{2} \left( 1 + \frac{\delta}{1 - 2\delta} \right) \left( 1 - \frac{1}{(1 - \delta)m} \right) + \frac{1}{2} \left( \frac{1}{(1 - \delta)m} - 1 \right) \tag{110}$$

It has been proven in case (i) that the first term $\left[ \frac{1}{2} \left( 1 + \frac{\delta}{1-2\delta} \right) \left( 1 - \frac{1}{(1-\delta)m} \right) \right]$ is monotonically increasing w.r.t. $\delta$. The second term $\frac{1}{2} \left( \frac{1}{(1-\delta)m} - 1 \right)$ is also monotonically increasing w.r.t. $\delta$. Thus, for the lower bound of $\kappa$, $\tilde{c}(\delta)$ is also monotonically increasing w.r.t. $\delta$.

In summary, under the definition of $(f, \kappa)$-robustness, the optimal batch size that minimize the theoretical upper bound for ByzSGDm and ByzSGDnm increase with the fraction of Byzantine workers $\delta$. The results are consistent with those under the definition of $(\delta_{\max}, c)$-robust aggregator.

# D COMPARISON WITH BYZ-VR-MARINA

We compare ByzSGDnm with Byz-VR-MARINA (Gorbunov et al., 2023) in this section.

## D.1 VARIANCE REDUCTION TECHNIQUES

We first compare the techniques for variance reduction in ByzSGDnm and Byz-VR-MARINA. As presented in the main text, in ByzSGDnm, the variance of stochastic gradients is reduced mainly by using (i) large batch size and (ii) local momentums. The two techniques introduce little extra computation cost but are empirically effective, as the empirical results in the main text show. However, the two techniques work only in the i.i.d. cases that we focus on in this work.

Byz-VR-MARINA adopts PAGE (Li et al., 2021) for variance reduction. In contrast to the techniques in ByzSGDnm, PAGE can reduce the bias in general non-i.i.d. cases but introduces much more computation cost (as we will shown in Appendix D.3).

## D.2 COMMUNICATION COST

ByzSGDnm and Byz-VR-MARINA take different ways to reduce the communication cost in distributed learning. ByzSGDnm uses large-batch training (Cutkosky & Mehta, 2020; Goyal et al.,

2017; You et al., 2020) to reduce the number communication round while Byz-VR-MARINA uses communication compression such as quantization (Alistarh et al., 2017) to reduce the transmitted bits in each communication round.

### D.3 COMPUTATION COST

Since the computation cost mainly comes from gradient computation in many real-world distributed learning applications, we use the number of gradient computation on non-Byzantine workers to estimate the computation cost for both methods. In ByzSGDnm, each of the $m(1 - \delta)$ non-Byzantine workers will draw a mini-batch of $B$ samples and compute the corresponding stochastic gradients at each iteration. Thus, the total number of gradient computation in each iteration for ByzSGDnm is $Bm(1 - \delta)$. In each iteration of Byz-VR-MARINA, each of the $m(1 - \delta)$ non-Byzantine workers computes the full gradient with probability $p$ and computes a mini-batch estimation of gradient differences with probability $1 - p$. Since the estimation with batch size $B$ requires $2B$ times of gradient computation, the expectation of the total number of gradient computation in each iteration for Byz-VR-MARINA is $(1 - \delta)np + 2Bm(1 - \delta)(1 - p)$ where $n$ is the total number of training instances. Therefore, the number of gradient computation in Byz-VR-MARINA for each iteration is $k_{\mathcal{C}}$ times that of ByzSGDnm in expectation, where

$$k_{\mathcal{C}} \triangleq \frac{(1 - \delta)np + 2Bm(1 - \delta)(1 - p)}{Bm(1 - \delta)} = \frac{np + 2Bm(1 - p)}{Bm} = 2 + p\left(\frac{n}{Bm} - 2\right).$$

Since $Bm$ is the total batch size of each iteration and $n$ is the total number of instances, $\frac{n}{Bm}$ is the number of iteration per epoch, which is typically much larger than 2. Therefore, the number of gradient computation for Byz-VR-MARINA is more than twice that of ByzSGDnm in expectation.

### D.4 THEORETICAL CONVERGENCE ORDER

As we have proved in Section 4 in the main text, under Assumptions 1, 2 and 3, we have

$$\min_{t=0,\ldots,T-1} \mathbb{E}\|\nabla F(\mathbf{w}_t)\| \leq O\left(\frac{1}{T^{\frac{1}{4}}}\right)$$

or equivalently

$$\min_{t=0,\ldots,T-1} \mathbb{E}\|\nabla F(\mathbf{w}_t)\|^2 \leq O\left(\frac{1}{T^{\frac{1}{2}}}\right)$$

for ByzSGDnm with $(\delta_{\max}, c)$-robust aggregators. Meanwhile, as the results in Gorbunov et al. (2023) show, under similar conditions, we have

$$\min_{t=0,\ldots,T-1} \mathbb{E}\|\nabla F(\mathbf{w}_t)\|^2 \leq O\left(\frac{1}{T}\right)$$

for Byz-VR-MARINA. It has been shown in existing works (Arjevani et al., 2023; Cutkosky & Mehta, 2020) that the convergence order $\min_t \mathbb{E}\|\nabla F(\mathbf{w}_t)\| \leq O(1/T^{\frac{1}{4}})$ is optimal for SGD under Assumptions 1, 2 and 3. Byz-VR-MARINA achieves the faster convergence order mainly because of intermittently using full gradients for variance reduction. The strategy of intermittently using full gradients is also adopted in some traditional methods such as SVRG (Johnson & Zhang, 2013). Although using full gradients improves the theoretical convergence order, it also increases the expected number of gradient computation as discussed in Appendix D.3 above.

In summary, compared to ByzSGDnm, Byz-VR-MARINA has a faster theoretical convergence order with respect to the iteration number, but also requires more times of gradient computation per iteration. Moreover, the number of gradient computation for Byz-VR-MARINA depends on $p$ and $n$.

### D.5 EMPIRICAL PERFORMANCE

Finally, we empirically compare the performance of ByzSGDnm and Byz-VR-MARINA. The experimental settings are the same as those presented in Section 5 of the main text. Specifically, we use cosine annealing (Loshchilov & Hutter, 2017) learning rates for each method. The initial learning rate for Byz-VR-MARINA is selected from $\{0.005, 0.01, 0.02, 0.05, 0.1, 0.2, 0.5, 1.0\}$, and

the best final top-1 test accuracy is used as the final metrics. We set the batch size to $512 \times 8$ for ByzSGDnm. We use the top-1 test accuracy w.r.t. gradient computation number as final metrics. For fairness, we do not use communication compression for Byz-VR-MARINA. The empirical results of Byz-VR-MARINA when the batch size is $32 \times 8$, $64 \times 8$, $128 \times 8$, $256 \times 8$ and $512 \times 8$ are presented in Figure 1, Figure 2, Figure 3, Figure 4 and Figure 5, respectively.

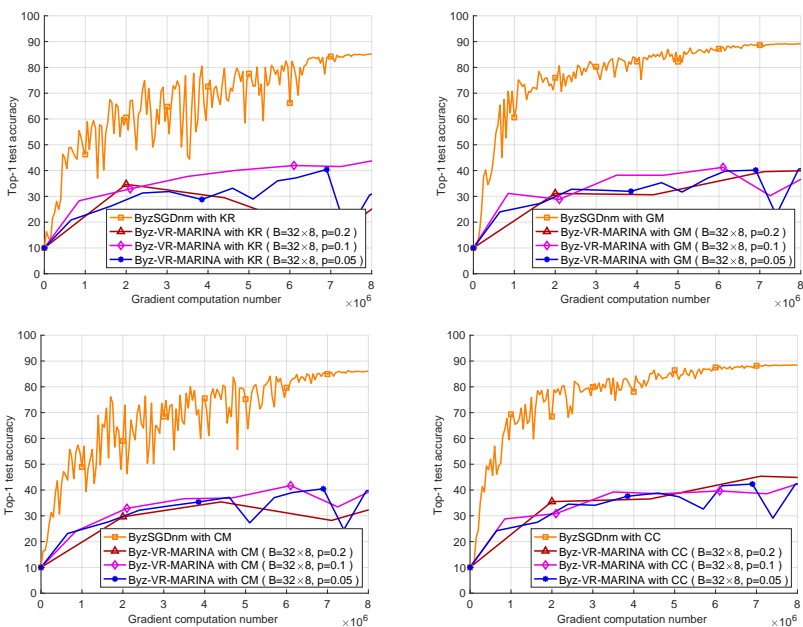

Figure 1: Top-1 test accuracy w.r.t. gradient computation number of different methods when there are 3 workers under ALIE attack. The batch size for Byz-VR-MARINA is set to $32 \times 8$.

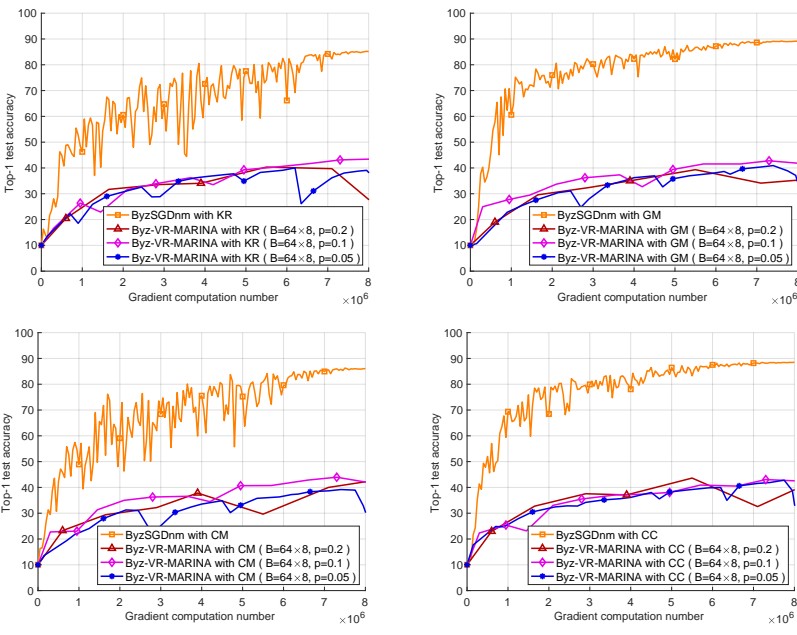

Figure 2: Top-1 test accuracy w.r.t. gradient computation number of different methods when there are 3 workers under ALIE attack. The batch size for Byz-VR-MARINA is set to $64 \times 8$.

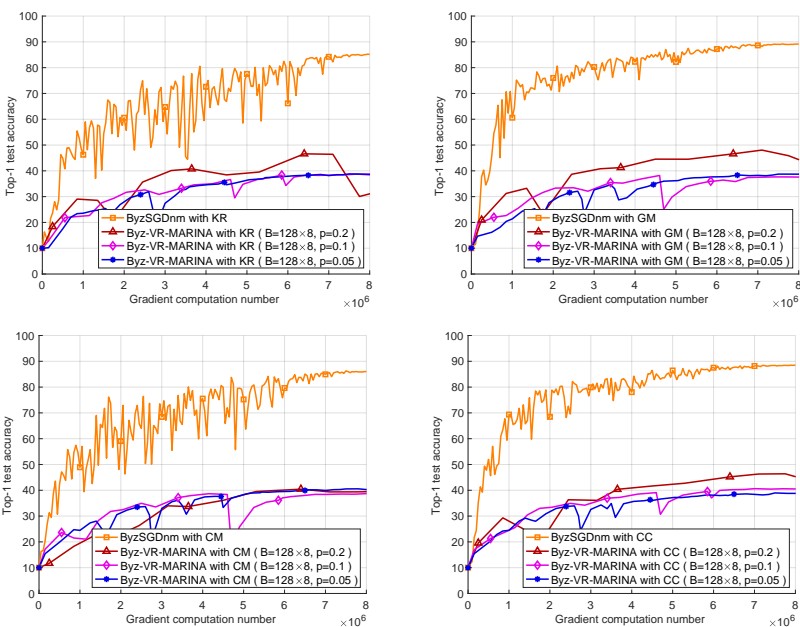

Figure 3: Top-1 test accuracy w.r.t. gradient computation number of different methods when there are 3 workers under ALIE attack. The batch size for Byz-VR-MARINA is set to $128 \times 8$.

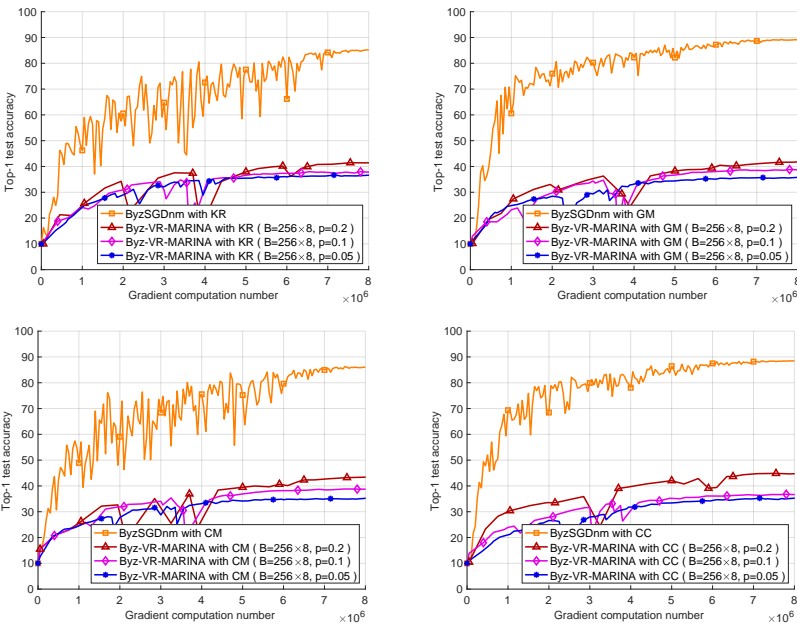

Figure 4: Top-1 test accuracy w.r.t. gradient computation number of different methods when there are 3 workers under ALIE attack. The batch size for Byz-VR-MARINA is set to $256 \times 8$.

As illustrated in Figure 1, Figure 2, Figure 3, Figure 4 and Figure 5, Byz-VR-MARINA converges much more slowly and has a much lower final top-1 accuracy than ByzSGDnm. There are mainly two reasons. Firstly, the full gradients in Byz-VR-MARINA is used to alleviate the bias in non-i.i.d. cases and is computation expensive. However, in i.i.d. cases that we focus on in this work, the full gradient is unnecessary and requires a large number times of gradient computation. Secondly, it

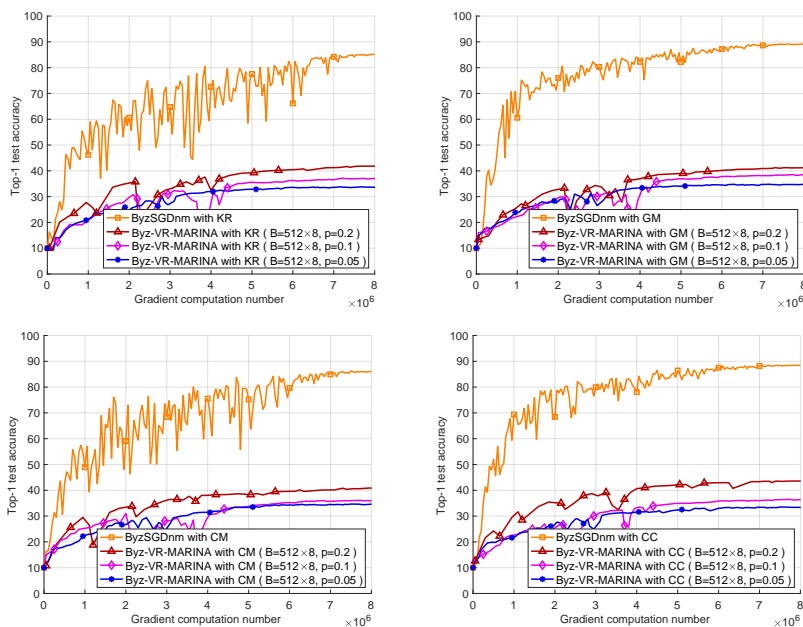

Figure 5: Top-1 test accuracy w.r.t. gradient computation number of different methods when there are 3 workers under ALIE attack. The batch size for Byz-VR-MARINA is set to $512 \times 8$.

has been shown in existing works (Defazio & Bottou, 2019) that using full gradients for variance reduction does not improve over SGD on deep learning models.

# E  MORE EXPERIMENTAL RESULTS

## E.1  INDEPENDENT AND IDENTICALLY DISTRIBUTED (I.I.D.) CASE

**Final top-1 test accuracy when using different batch size.** We present the empirical results of ByzSGDm and ByzSGDnm with different batch size (ranging from $32 \times 8$ to $1024 \times 8$) under no attack or failure, under bit-flipping failure (Xie et al., 2019), ALIE attack and FoE attack in Table 7, Table 8, Table 9 and Table 10, respectively. Specifically, workers under bit-flipping failure will send the vectors that are $-10$ times the true values. The final top-1 test accuracy of the two methods under ALIE attack when NNM technique (Allouah et al., 2023) is used is presented in Table 11 below.

Table 7: The final top-1 test accuracy of ByzSGDm and ByzSGDnm with different batch size when there is no attack or failure

| Batch size | $32 \times 8$ | $64 \times 8$ | $128 \times 8$ | $256 \times 8$ | $512 \times 8$ | $1024 \times 8$ |
|---|---|---|---|---|---|---|
| ByzSGDm + KR | **91.08%** | 89.98% | 89.71% | 89.15% | 86.15% | 84.97% |
| ByzSGDnm + KR | **91.00%** | 90.15% | 89.23% | 88.76% | 87.83% | 84.71% |
| ByzSGDm + GM | **92.02%** | 91.50% | 90.85% | 89.26% | 88.21% | 86.52% |
| ByzSGDnm + GM | **92.18%** | 91.81% | 91.22% | 89.93% | 90.01% | 88.08% |
| ByzSGDm + CM | **92.30%** | 91.79% | 90.43% | 89.84% | 87.27% | 84.06% |
| ByzSGDnm + CM | **92.29%** | 91.70% | 91.15% | 90.20% | 89.06% | 88.11% |
| ByzSGDm + CC | **92.52%** | 91.74% | 90.63% | 89.40% | 88.78% | 85.50% |
| ByzSGDnm + CC | **92.51%** | 91.91% | 91.50% | 90.00% | 89.33% | 88.47% |

Table 8: The final top-1 test accuracy of ByzSGDm and ByzSGDnm with different batch size when there are 3 Byzantine workers under bit-flipping failure

| Batch size | 32×8 | 64×8 | 128×8 | 256×8 | 512×8 | 1024×8 |
|---|---|---|---|---|---|---|
| ByzSGDm + KR | **91.09%** | 90.30% | 89.55% | 88.37% | 87.52% | 85.68% |
| ByzSGDnm + KR | **90.71%** | 90.14% | 89.59% | 88.89% | 85.76% | 82.22% |
| ByzSGDm + GM | 88.97% | **89.18%** | 88.61% | 87.43% | 85.72% | 83.56% |
| ByzSGDnm + GM | 88.64% | **89.16%** | 88.89% | 88.22% | 87.78% | 86.21% |
| ByzSGDm + CM | 86.80% | 87.11% | **87.40%** | 86.37% | 85.40% | 81.23% |
| ByzSGDnm + CM | 87.39% | **88.12%** | 87.66% | 86.76% | 86.33% | 85.52% |
| ByzSGDm + CC | 88.92% | **88.97%** | 88.78% | 88.02% | 86.54% | 83.93% |
| ByzSGDnm + CC | 88.81% | 88.89% | **88.96%** | 88.45% | 87.56% | 85.53% |

Table 9: The final top-1 test accuracy of ByzSGDm and ByzSGDnm with different batch size when there are 3 Byzantine workers under ALIE attack

| Batch size | 32×8 | 64×8 | 128×8 | 256×8 | 512×8 | 1024×8 |
|---|---|---|---|---|---|---|
| ByzSGDm + KR | 38.55% | 54.15% | 55.98% | 59.28% | 83.42% | **83.45%** |
| ByzSGDnm + KR | 43.47% | 70.88% | 80.20% | 82.83% | 85.12% | **85.93%** |
| ByzSGDm + GM | 63.11% | 70.88% | 82.08% | **87.62%** | 86.95% | 84.75% |
| ByzSGDnm + GM | 69.45% | 83.23% | 86.63% | 88.66% | **89.13%** | 88.16% |
| ByzSGDm + CM | 33.11% | 55.66% | 66.38% | 82.47% | **83.25%** | 80.94% |
| ByzSGDnm + CM | 61.28% | 71.46% | 80.24% | 83.55% | **86.03%** | 85.74% |
| ByzSGDm + CC | 72.83% | 79.45% | 84.94% | 87.25% | **87.46%** | 83.70% |
| ByzSGDnm + CC | 78.50% | 83.91% | 86.56% | 88.32% | **88.53%** | 87.89% |

Table 10: The final top-1 test accuracy of ByzSGDm and ByzSGDnm with different batch size when there are 3 Byzantine workers under FoE attack

| Batch size | 32×8 | 64×8 | 128×8 | 256×8 | 512×8 | 1024×8 |
|---|---|---|---|---|---|---|
| ByzSGDm + KR | 10.00% | 10.00% | 10.00% | 10.00% | 10.00% | 10.00% |
| ByzSGDnm + KR | 10.00% | 10.00% | 10.00% | 10.00% | 10.00% | 10.00% |
| ByzSGDm + GM | 78.36% | 81.98% | 82.69% | 82.20% | **84.09%** | 78.90% |
| ByzSGDnm + GM | 88.55% | 88.75% | **90.99%** | 90.23% | 89.12% | 88.38% |
| ByzSGDm + CM | 83.97% | **84.28%** | 84.01% | 83.48% | 79.16% | 78.76% |
| ByzSGDnm + CM | 84.12% | 84.77% | 85.23% | **85.74%** | 84.65% | 83.36% |
| ByzSGDm + CC | 83.60% | 84.26% | 87.45% | **88.48%** | 86.24% | 81.36% |
| ByzSGDnm + CC | 88.99% | 90.07% | **90.69%** | 90.54% | 89.32% | 88.20% |

Table 11: The final top-1 test accuracy of ByzSGDm and ByzSGDnm with different batch size when there are 3 Byzantine workers under ALIE attack and NNM technique is used

| Batch size | 32×8 | 64×8 | 128×8 | 256×8 | 512×8 | 1024×8 |
|---|---|---|---|---|---|---|
| ByzSGDm + KR | 58.61% | 65.96% | 78.37% | **85.71%** | 85.26% | 83.97% |
| ByzSGDnm + KR | 80.41% | 83.85% | 85.88% | 87.15% | **87.68%** | 87.09% |
| ByzSGDm + GM | 72.58% | 73.64% | 78.02% | **85.73%** | 85.37% | 85.32% |
| ByzSGDnm + GM | 79.50% | 83.96% | 86.42% | 86.91% | **88.09%** | 87.23% |
| ByzSGDm + CM | 71.51% | 78.15% | 82.95% | 86.06% | **86.95%** | 85.24% |
| ByzSGDnm + CM | 79.81% | 84.14% | 85.69% | 87.65% | **87.69%** | 87.11% |
| ByzSGDm + CC | 76.48% | 81.18% | 84.63% | **86.65%** | 85.98% | 85.36% |
| ByzSGDnm + CC | 79.91% | 83.50% | 87.00% | 87.48% | 87.59% | **87.78%** |

**More results about wall-clock time.** In Section 5 in the main text, we have reported the wall-clock time of different methods for 160 epochs, which empirically shows that using large batch size has the bonus of training acceleration. To further support the conclusion, we also present the wall-clock time that different methods require to reach 50%, 75% and 85% top-1 test accuracy in Table 12, Table 13 and Table 14, respectively. As we can see from the results, ByzSGDnm requires less time to reach the target accuracy than ByzSGDm in most cases. Moreover, setting a relatively large batch size can significantly accelerate the training process.

Table 12: The wall-clock time for different methods to reach 50% top-1 test accuracy when there are 3 Byzantine workers under ALIE attack. The missing values mean that the target test accuracy is not reached in 160 epochs for the corresponding methods.

| Batch size | $32 \times 8$ | $64 \times 8$ | $128 \times 8$ | $256 \times 8$ | $512 \times 8$ | $1024 \times 8$ |
|---|---|---|---|---|---|---|
| ByzSGDm + KR | – | 157.10s | 34.04s | 33.93s | 31.32s | 70.77s |
| ByzSGDnm + KR | 408.89s | 31.46s | 26.27s | 27.66s | 33.35s | 44.31s |
| ByzSGDm + GM | 74.21s | 42.70s | 26.27s | 28.90s | 38.77s | 55.85s |
| ByzSGDnm + GM | 61.09s | 24.92s | 17.43s | 24.18s | 23.24s | 38.24s |
| ByzSGDm + CM | 303.55s | 110.69s | 28.52s | 37.20s | 36.52s | 73.44s |
| ByzSGDnm + CM | 475.48s | 41.17s | 22.32s | 27.35s | 27.16s | 45.66s |
| ByzSGDm + CC | 102.17s | 37.60s | 17.89s | 29.08s | 27.11s | 67.76s |
| ByzSGDnm + CC | 63.56s | 25.20s | 20.75s | 10.86s | 21.45s | 27.22s |

Table 13: The wall-clock time for different methods to reach 75% top-1 test accuracy when there are 3 Byzantine workers under ALIE attack. The missing values mean that the target test accuracy is not reached in 160 epochs for the corresponding methods.

| Batch size | $32 \times 8$ | $64 \times 8$ | $128 \times 8$ | $256 \times 8$ | $512 \times 8$ | $1024 \times 8$ |
|---|---|---|---|---|---|---|
| ByzSGDm + KR | – | – | – | – | 133.69s | 148.70s |
| ByzSGDnm + KR | – | – | 328.49s | 123.12s | 130.45s | 134.09s |
| ByzSGDm + GM | – | – | 114.38s | 59.19s | 90.74s | 124.03s |
| ByzSGDnm + GM | – | 252.19s | 66.13s | 54.12s | 45.07s | 76.49s |
| ByzSGDm + CM | – | – | – | 202.97s | 110.25s | 204.42s |
| ByzSGDnm + CM | – | – | 322.83s | 150.25s | 63.85s | 109.23s |
| ByzSGDm + CC | – | 727.07s | 77.19s | 87.42s | 86.53s | 150.59s |
| ByzSGDnm + CC | 1465.07s | 112.49s | 109.18s | 56.20s | 61.30s | 81.28s |

Table 14: The wall-clock time for different methods to reach 85% top-1 test accuracy when there are 3 Byzantine workers under ALIE attack. The missing values mean that the target test accuracy is not reached in 160 epochs for the corresponding methods.

| Batch size | $32 \times 8$ | $64 \times 8$ | $128 \times 8$ | $256 \times 8$ | $512 \times 8$ | $1024 \times 8$ |
|---|---|---|---|---|---|---|
| ByzSGDm + KR | – | – | – | – | – | – |
| ByzSGDnm + KR | – | – | – | – | 279.88s | 248.95s |
| ByzSGDm + GM | – | – | – | 269.31s | 218.80s | – |
| ByzSGDnm + GM | – | – | 408.89s | 201.76s | 176.06s | 171.29s |
| ByzSGDm + CM | – | – | – | – | – | – |
| ByzSGDnm + CM | – | – | – | – | 240.10s | 257.50s |
| ByzSGDm + CC | – | – | – | 292.38s | 207.57s | – |
| ByzSGDnm + CC | – | – | 426.95s | 216.13s | 176.83s | 188.12s |

**Top-1 test accuracy w.r.t. gradient computation number / wall-clock time.** We present the top-1 test accuracy w.r.t. gradient computation number and wall-clock time for different methods when there are 3 workers under ALIE attack in Figure 6 and Figure 7, respectively. The empirical results show that using a relatively large batch size ($B = 512 \times 8$) can lead to a stabler training process and higher final top-1 test accuracy. Moreover, under each setting of aggregators and batch size in the experiment, ByzSGDnm outperforms ByzSGDm in final top-1 test accuracy.

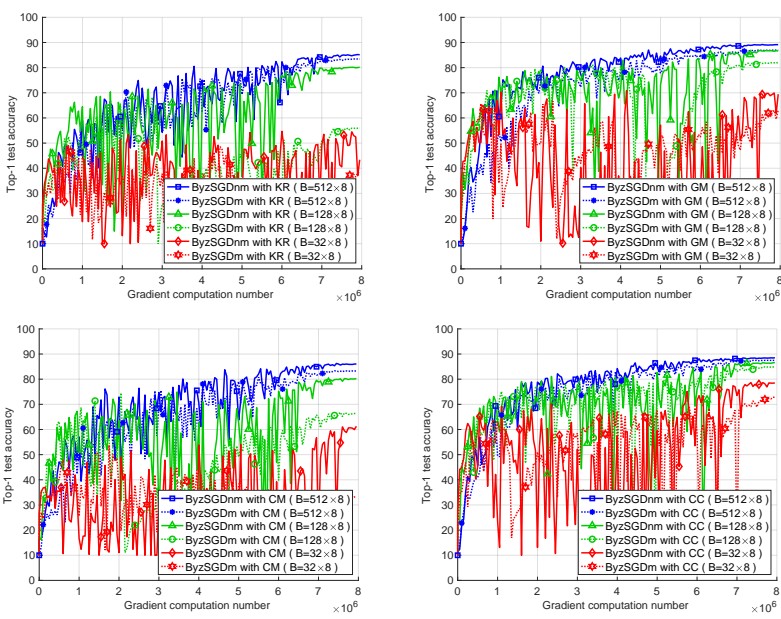

Figure 6: Top-1 test accuracy w.r.t. gradient computation number of ByzSGDnm and ByzSGDm with different batch size when there are 3 workers under ALIE attack

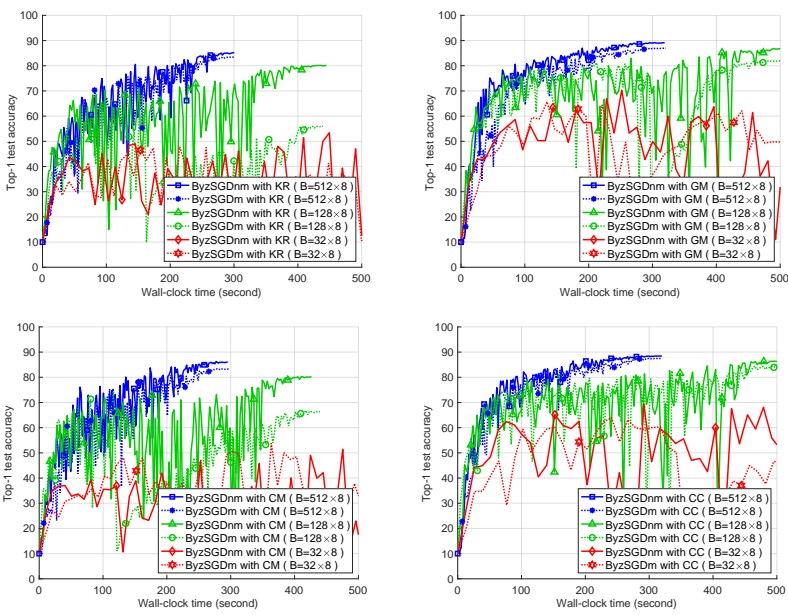

Figure 7: Top-1 test accuracy w.r.t. wall-clock time of ByzSGDnm and ByzSGDm with different batch size when there are 3 workers under ALIE attack

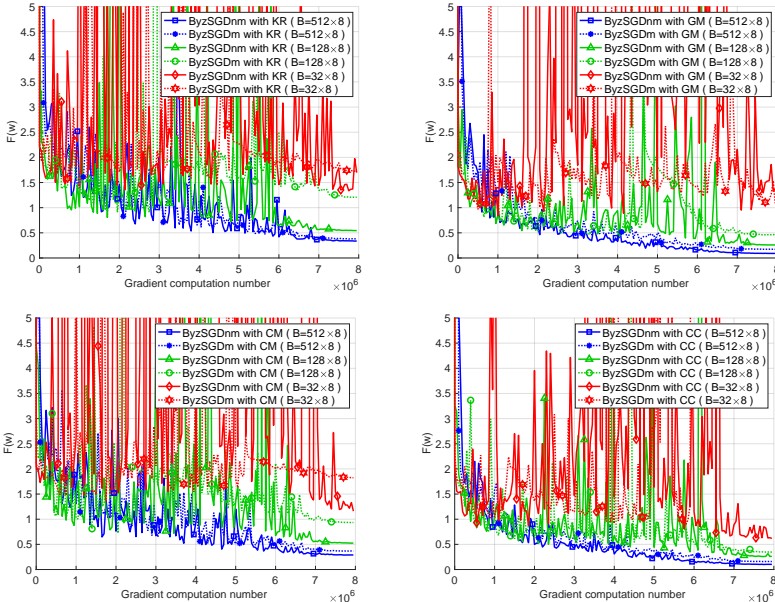

Figure 8: Training loss $F(\mathbf{w})$ w.r.t. gradient computation number of ByzSGDnm and ByzSGDm with different batch size when there are 3 workers under ALIE attack

Meanwhile, we also present the training loss w.r.t. gradient computation number in Figure 8. The results further verify the effectiveness of large batch size and ByzSGDnm.

### E.2 Non-i.i.d. Case

Although we mainly focus on the i.i.d. case in this paper, we also provide some empirical results in non-i.i.d. cases in this section. Specifically, we randomly sample from the training set of CIFAR-10 dataset according to the Dirichlet distribution with hyper-parameter 1.0. The number of training instances for each class on each worker is presented in Table E.1 below.

Table 15: The number of training instances for each class on each worker

| Class label | 0 | 1 | 2 | 3 | 4 | 5 | 6 | 7 | 8 | 9 | Total |
|---|---|---|---|---|---|---|---|---|---|---|---|
| Worker 0 | 523 | 334 | 62 | 582 | 491 | 2502 | 721 | 148 | 568 | 319 | 6250 |
| Worker 1 | 492 | 898 | 697 | 159 | 83 | 92 | 787 | 2415 | 67 | 560 | 6250 |
| Worker 2 | 754 | 465 | 459 | 426 | 2365 | 167 | 121 | 815 | 678 | 0 | 6250 |
| Worker 3 | 0 | 0 | 61 | 159 | 749 | 304 | 364 | 671 | 688 | 3254 | 6250 |
| Worker 4 | 1106 | 515 | 1692 | 652 | 593 | 611 | 553 | 70 | 0 | 458 | 6250 |
| Worker 5 | 2004 | 105 | 105 | 2608 | 29 | 0 | 0 | 345 | 1054 | 0 | 6250 |
| Worker 6 | 59 | 2146 | 1561 | 314 | 564 | 395 | 324 | 135 | 752 | 0 | 6250 |
| Worker 7 | 62 | 537 | 363 | 100 | 126 | 929 | 2130 | 401 | 1193 | 409 | 6250 |
| Total | 5000 | 5000 | 5000 | 5000 | 5000 | 5000 | 5000 | 5000 | 5000 | 5000 | 50000 |

Moreover, we replace the batch normalization layers in the ResNet-20 deep learning model (He et al., 2016) with group normalization layers (Wu & He, 2018) as suggested in existing works (Hsieh et al., 2020). We empirically compare ByzSGDnm with ByzSGDm and Byz-VR-MARINA (Gorbunov et al., 2023). We try the batch size $32 \times 8$, $128 \times 8$ and $512 \times 8$ for each method. Among the 8 workers, one worker (worker 0) is under ALIE attack. The other hyper-parameter settings for ByzSGDm and ByzSGDnm are the same as those in Section 5 of the main text. The other hyper-parameter settings for Byz-VR-MARINA are the same as those in Appendix D.5.

**Comparison among the methods in the non-i.i.d. case.** The top-1 test accuracy w.r.t. gradient computation number is illustrated in Figure 9. We also present the final top-1 test accuracy of ByzSGDm and ByzSGDnm with different batch size in Table 16. As the empirical results show, ByzSGDnm still outperforms ByzSGDm and Byz-VR-MARINA under this non-i.i.d. setting. Moreover, for ByzSGDnm and ByzSGDm, increasing batch size can still increase top-1 test accuracy under this non-i.i.d. setting.

**Further comparison when NNM technique is used.** We also test the empirical performance of the three methods (ByzSGDnm, ByzSGDm and Byz-VR-MARINA) under the non-i.i.d. setting when nearest neighbour mixing (NNM) (Allouah et al., 2023) technique is used. The top-1 test accuracy w.r.t. gradient computation number when there is 1 worker under ALIE attack is illustrated in Figure 10. We also present the final top-1 test accuracy of ByzSGDm and ByzSGDnm with different batch size in Table 17. Compared to the case without NNM, the final top-1 accuracy of ByzSGDnm and ByzSGDm significantly increases (from about 35% to about 80%) when using NNM technique. On the contrary, the final top-1 test accuracy of Byz-VR-MARINA is still around 20% when NNM is used. In addition, for ByzSGDm and ByzSGDnm, the batch size that leads to the best top-1 test accuracy decreases when combined with NNM. A possible reason is that when combined with NNM, the term $c$ in equation (4) decreases, leading to the decrease of $B^*$. However, since the bias should also be taken into consideration, it requires further work to study the effect of batch size in non-i.i.d. cases.

Meanwhile, as the results in Table 17 and in Figure 10 show, ByzSGDnm still outperforms ByzSGDm when combined with NNM in non-i.i.d. cases. The empirical results show that in non-i.i.d. cases, ByzSGDnm is still a promising choice. Since we mainly focus on the i.i.d. case in this work, we will further study the behavior of ByzSGDnm in non-i.i.d. cases in future work.

Table 16: The final top-1 test accuracy of ByzSGDm and ByzSGDnm with GM in the non-i.i.d. setting when there is 1 Byzantine worker under ALIE attack.

| Batch size | 32×8 | 64×8 | 128×8 | 256×8 | 512×8 | 1024×8 |
|---|---|---|---|---|---|---|
| ByzSGDm | 10.16% | 23.33% | 27.78% | 29.63% | **32.90%** | 32.53% |
| ByzSGDnm | 23.87% | 26.78% | 28.42% | 30.04% | 35.24% | **36.25%** |

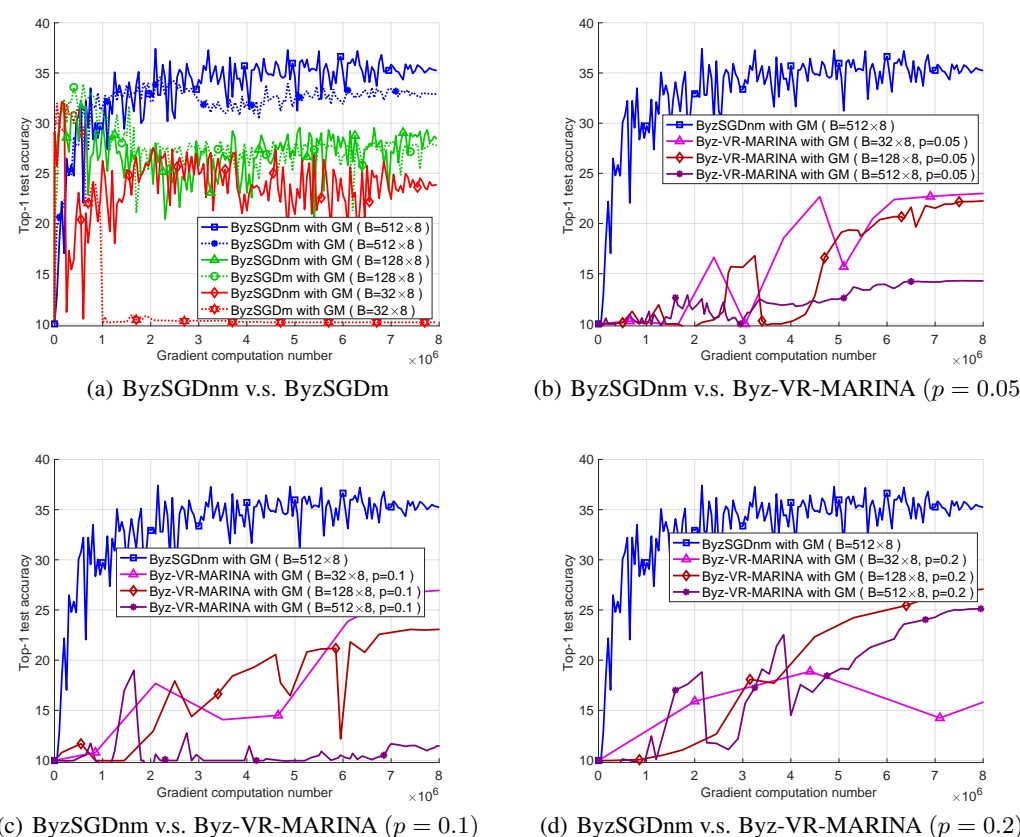

(a) ByzSGDnm v.s. ByzSGDm

(b) ByzSGDnm v.s. Byz-VR-MARINA ($p = 0.05$)

(c) ByzSGDnm v.s. Byz-VR-MARINA ($p = 0.1$)

(d) ByzSGDnm v.s. Byz-VR-MARINA ($p = 0.2$)

Figure 9: Top-1 test accuracy w.r.t. gradient computation number when there is 1 worker under ALIE attack under the non-i.i.d. setting

Table 17: The final top-1 test accuracy of ByzSGDm and ByzSGDnm with GM in the non-i.i.d. setting when there is 1 Byzantine worker under ALIE attack and NNM technique is used.

| Batch size | 16×8 | 32×8 | 64×8 | 128×8 | 256×8 | 512×8 |
|---|---|---|---|---|---|---|
| ByzSGDm | 80.46% | **80.56%** | 78.29% | 74.10% | 66.71% | 58.78% |
| ByzSGDnm | 81.11% | **81.74%** | 80.08% | 77.98% | 74.65% | 69.05% |

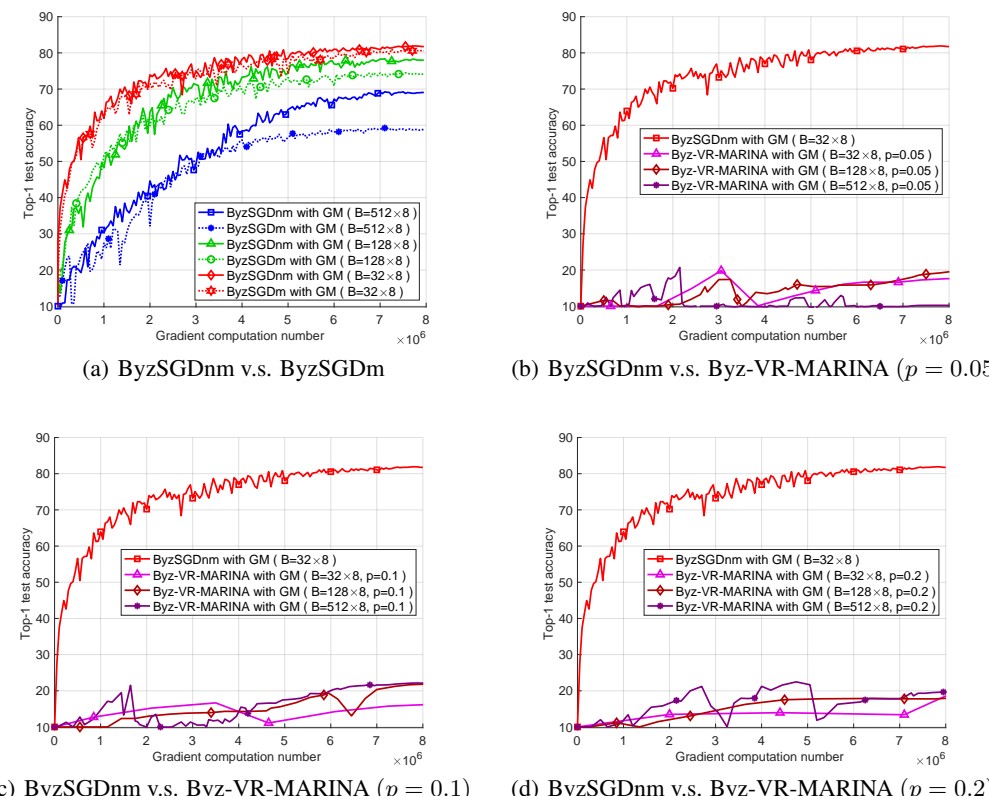

(a) ByzSGDnm v.s. ByzSGDm

(b) ByzSGDnm v.s. Byz-VR-MARINA ($p = 0.05$)

(c) ByzSGDnm v.s. Byz-VR-MARINA ($p = 0.1$)

(d) ByzSGDnm v.s. Byz-VR-MARINA ($p = 0.2$)

Figure 10: Top-1 test accuracy w.r.t. gradient computation number when there is 1 worker under ALIE attack under the non-i.i.d. setting and NNM technique is used

