# OpenReview forum: "On the Effect of Batch Size in Byzantine-Robust Distributed Learning"
_ICLR.cc/2024/Conference — ICLR 2024 poster_

### Official Review · Reviewer_tTch · 2023-10-30

**Soundness:** 4 excellent
**Presentation:** 4 excellent
**Contribution:** 3 good
**Rating:** 6
**Confidence:** 5

**Summary:**

This work studies Byzantine-robust distributed learning for the i.i.d. non-convex smooth case. The paper proposes two tricks to improve the existing methods, i.e., large batch size and normalized momentum. The authors provide theoretical arguments to show the benefits of large batch size (variance reduction) and prove the convergence of normalized momentum trick. Empirical experiments show that the combination of these two tricks outperforms existing start-of-the-art methods.

**Strengths:**

1. The proposed normalized momentum shows significant empirical improvement in the small batch size case as shown in table 3 and table 4.
2. The use of large batch size also significantly boosts the test accuracy under ALIE attack and FoE attack except for KR and CM for the FoE attack cases.
3. The combination of large batch size and normalized momentum achieves the best empirical performance in nearly every case.
4. The use of large batch size significantly reduce the wall-clock running time for training fixed number of epochs.

**Weaknesses:**

1. There exist little theoretical improvements regarding to existing BRDL methods in terms of problem assumptions or convergence rates. In addition, this work only considers i.i.d. cases, which is kind of restrictive if not enough theoretical improvements are obtained.
2. The variance reduction trick using large batch size is a direct consequence of (Karimireddy et al., 2021)), so it is hard to claim that this is one of the main contributions of the current work. Furthermore, the optimization of B is conducted on the upper bounds of the convergence rates, so it does not necessarily leads to faster convergence if we set optimal B. This probably should be made clear in the paper.
3. The technical elements in proving the convergence of ByzSGDnm are closely related to references such as (Cutkosky & Mehta, 2020), so I am not clear whether there are substantial contributions therein.

**Questions:**

1. SGDm is not defined before first used.
2. In proposition 2, equation (5), why the last term has a $\sigma^2$ term, while in the work (Cutkosky & Mehta, 2020), this term is only in $\sigma$. Can you briefly explain why?
3. Can you explain why KR fails all cases? Is that because KR does not satisfy the definition 1?
4. Why does CM have degraded performance in table 4 after increasing batch size, this batch size should be right based on its performance in ByzSGDnm, can you provide some comments on that?
5. In the comparison with Byz-VR-MARINA, why do you use $512\times 8$ batch size for ByzSGDnm, but never uses that for Byz-VR-MARINA?

---

> ### Author Response · Authors · 2023-11-21
> **Official Comment by Authors (part 2/2)**
>
> **Question 2. In proposition 2, equation (5), why the last term has a $\sigma^2$ term, while in the work (Cutkosky & Mehta, 2020), this term is only in $\sigma$. Can you briefly explain why?**
>
> After checking our proof and the proof in (Cutkosky & Mehta, 2020), we carefully guess that there might be a typo in the proof of (Cutkosky & Mehta, 2020). It seems that the term $\frac{8\sigma\sqrt{T}}{\sqrt{RL}}=O(\sigma)$ in the last step of the proof of Theorem 1 in (Cutkosky & Mehta, 2020) should be $\frac{8\sigma^2\sqrt{T}}{\sqrt{RL}}=O(\sigma^2)$ since it is obtained by letting $\alpha=\frac{\sqrt{RL}}{\sigma\sqrt{T}}$ in the term $\frac{8\sigma}{\alpha}$.
>
> Meanwhile, we are willing to hear from the reviewer if they have any further comments about this question.
>
> ---
>
> **Question 3. Can you explain why KR fails all cases? Is that because KR does not satisfy the definition 1?**
>
> We would like to point out that in the experiments of previous works (Karimireddy et al., 2021; Xie et al., 2020), KR also failed under FoE attack in all cases. To the best of our knowledge, it is unknown whether KR satisfies Definition 1 or not. Although we do not focus on the performance of a specific aggregator under a specific attack in this work, we are willing to provide some comments on this phenomenon below.
>
> Firstly, the theoretical guarantee for Krum (KR) in (Blanchard et al., 2017) requires that $2f+2<n$ where $f$ is the number of Byzantine workers and $n$ is the number of all workers. The condition is not satisfied in our experiment where $3$ workers among $8$ workers are Byzantine. In addition, we find that when using KR under FoE attack, the model parameter falls into an area with a small gradient norm but a large training loss. Since theoretical results only guarantee a small gradient norm in expectation, the poor empirical performance of KR does not conflict with the theoretical results.
>
> ---
>
> **Question 4. Why does CM have degraded performance in table 4 after increasing batch size?**
> As the detailed results in Appendix E.1 show, [ByzSGDm + CM] has the best performance when $B=128\times 8$ in this case. When $B<128\times8$, the performance of [ByzSGDm + CM] improves as $B$ increases. When $B>128\times8$, the performance of [ByzSGDm + CM] degrades as $B$ increases. It is consistent with our theoretical results and other empirical results.
>
> For quick access, we present part of the results in Table 10 in the revised version below, which show the top-$1$ test accuracy of different methods when there are $3$ workers under FoE attack.
>
> |Batch size|32$\times$8 | 64$\times$8 | 128$\times$8 | 256$\times$8 | 512$\times$8| 1024$\times$8|
> |----|----|----|----|----|----|----|
> |ByzSGDm + GM | 78.36\% | 81.98\% | 82.69\% | 82.20\% | **84.09\%** | 78.90\% |
> |ByzSGDnm + GM | 88.55\% | 88.75\% | **90.99\%** | 90.23\% | 89.12\% | 88.38\% |
> |ByzSGDm + CM | 83.97\% | **84.28\%** | 84.01\% | 83.48\% | 79.16\% | 78.76\% |
> |ByzSGDnm + CM | 84.12\% | 84.77\% | 85.23\% | **85.74\%** | 84.65\% | 83.36\% |
> |ByzSGDm + CC | 83.60\% | 84.26\% | 87.45\% | **88.48\%** | 86.24\% | 81.36\% |
> |ByzSGDnm + CC | 88.99\% | 90.07\% | **90.69\%** | 90.54\% | 89.32\% | 88.20\% |
>
> ---
>
> **Question 5. In the comparison with Byz-VR-MARINA, why do you use batch size $512\times 8$ for ByzSGDnm, but never uses that for Byz-VR-MARINA?**
>
> This is mainly because Byz-VR-MARINA adopts the PAGE style, which prefers a relatively small batch size (Li et al., 2021, Gorbunov et al., 2023). Meanwhile, we have added the empirical results of Byz-VR-MARINA when $B=256\times 8$ and $B=512\times 8$ in Appendix D.5 in the revised version. The empirical results show that the performance of Byz-VR-MARINA does not improve when the batch size increases to $256\times 8$ or $512\times 8$.
>
> ---
>
> **Added empirical results for non-i.i.d. cases**
>
> We have also followed the other reviewers' suggestions and added the empirical results in a heterogeneous setup in Appendix E.2, which we think can improve the contribution of our work. In the added experiments, the training instances on workers are sampled by using a Dirichlet distribution with hyper-parameter $1.0$.
>
> The empirical results show that under this non-i.i.d. setting, using a relatively large batch size can still lead to higher test accuracy. Moreover, ByzSGDnm with batch size $512\times 8$ has the best final top-$1$ test accuracy.
> In addition, we have also tested the empirical performance of the three methods (ByzSGDnm, ByzSGDm, Byz-VR-MARINA) when combined with nearest neighbor mixing (NNM) technique under the non-i.i.d. setting. ByzSGDnm still has the best final top-$1$ test accuracy among the methods. Please refer to Appendix E.2. for more detailed results.
>
> ---
>
> We hope that we have addressed the reviewer's concerns, and we are always willing to answer any further questions. Meanwhile, we would greatly appreciate it if the reviewer could re-evaluate our work in light of our response.

---

> ### Author Response · Authors · 2023-11-21
> **Official Comment by Authors (part 1/2)**
>
> We sincerely thank the reviewer for the insightful comments. We have revised our work according to the suggestions and updated the revised version. The changed or added parts are marked in blue for quick recognition. Then, we respond to the raised concerns and questions point by point below.
>
> ---
>
> **Weakness 1. There exist little theoretical improvements regarding to existing BRDL methods in terms of problem assumptions or convergence rates.**
>
> We would like to clarify that the main theoretical contribution of our work lies in the analysis of the effect of batch size in BRDL. Specifically, as far as we know, we are the first to show that a relatively large batch size is preferred in BRDL (please see our response to Weakness 2(a)).
>
> In addition, we would like to point out that although in the same order, the convergence rate in our work is tighter than those in existing works for SGD-based methods in BRDL, as far as we know. Moreover, it has been shown in (Arjevani et al., 2019) that under Assumptions 1, 2, and 3 (please see Section 2), the convergence order $\mathbb{E}||\nabla F(w_T)||\leq O(1/T^\frac{1}{4})$ is optimal for SGD in the worst case. Therefore, we carefully think that the improvement by optimizing batch size in our work is meaningful.
>
> ---
>
> **Weakness 2(a). The variance reduction trick using large batch size is a direct consequence of (Karimireddy et al., 2021), so it is hard to claim that this is one of the main contributions of the current work.**
>
> We thank the reviewer for the comment. Actually, we would like to politely clarify that the results of (Karimireddy et al., 2021) do not suggest using large batch sizes. Although increasing batch size has the positive effect of reducing the variance of each iteration, it also has the negative effect of increasing the gradient computation number for each iteration. Therefore, for a given total gradient computation number $\mathcal{C}$, it is uncertain whether a large or small batch size is better in BRDL based only on the results (Karimireddy et al., 2021) without further analysis.
>
> In traditional distributed learning (TDL) without attacks, the variance of stochastic gradients will also slow down the convergence. However, relatively small batch size is still preferred in TDL since it allows more iterations and leads to better empirical performance when $\mathcal{C}$ is fixed (or equivalently, when the total epoch number is fixed). As far as we know, almost all existing works on BRDL  directly follow the setting of small batch sizes in TDL. However, our theoretical results show that a relatively large batch size is preferred in BRDL, which is contrary to that in TDL without attacks. Moreover, our theoretical findings are supported by empirical results.
>
> In summary, in this work, we have pointed out the improperly inherited setting of small batch size in BRDL by both theoretical analysis and empirical results. We would greatly appreciate it if the reviewer could re-evaluate the contribution of our work in light of our response above.
>
> ---
>
> **Weakness 2(b). The optimization of B is conducted on the upper bounds of the convergence rates, so it does not necessarily lead to faster convergence if we set optimal B. This probably should be made clear in the paper.**
>
> We thank the reviewer for the insightful comment. We have added the following statement at the end of Section 3.1, which is marked in blue.
>
> "In addition, we would like to clarify that although we obtain a better worst-case guarantee by optimizing $\mathcal{U}(B)$, it does not necessarily ensure a better empirical performance given the complexity and variety in real-world applications."
>
> ---
>
> **Weakness 3. The technical elements in proving the convergence of ByzSGDnm are closely related to references such as (Cutkosky & Mehta, 2020), so I am not clear whether there are substantial contributions therein.**
>
> We would like to point out that the theoretical results in our work cannot be obtained by simply combining the proof in (Cutkosky & Mehta, 2020) and (Karimireddy et al., 2021). Specifically, the term $(\frac{1}{\alpha T}+\sqrt{\alpha})$ in Theorem 2 cannot be obtained by directly following the proof in (Karimireddy et al., 2021). To overcome the problem, we provide a finer analysis for $\mathbb{E}||u_t-\bar{u}_t||^2$ in Lemma 2 (equation (29)). The term $[\alpha + (1-\alpha)^{2t}]$ is originally $[\alpha + (1-\alpha)^t]$ in Lemma 9 in Appendix E of (Karimireddy et al., 2021).
>
> ---
>
> **Question 1. SGDm is not defined before first used.**
>
> We thank the reviewer for pointing out the undefined abbreviation. It has been modified to 'stochastic gradient descent with momentum (SGDm)' in the revised version.

---

> > ### Comment · Reviewer_tTch · 2023-11-23
> >
> > I thank the reviewer for detailed answers to my questions and comments. I agree that in terms of the upper bound derived in the paper (Karimireddy et al., 2021), increasing the batch size as the fraction of Byzantine clients grows is beneficial and this is the first time it is discovered. However, I still think that optimizing an existing upper bound does not constitute substantial theoretical contribution, likewise, although the authors provide convergence analysis for the proposed method ByzSGDnm, but there is no theoretical improvements claimed in the paper compared to existing results, such as https://arxiv.org/pdf/2006.09365.pdf by setting heterogeneity to be zero. However, I agree that the large batch size trick and the proposed algorithm perform well in real experiments, and thus I would not object if the AC strongly supports this work.

---

> > > ### Author Response · Authors · 2023-11-23
> > >
> > > We sincerely thank the reviewer for acknowledging the novelty of our work. Meanwhile, we respectfully disagree with the reviewer and believe that there are adequate contributions in our work due to the following reasons:
> > >
> > > 1. As the reviewer acknowledges, we are the first to show that increasing the batch size as the fraction of Byzantine clients grows leads to a tighter upper bound (Proposition 1). The theoretical result suggests using a relatively large batch size in BRDL. Since almost all previous works on BRDL inherit the setting of small batch sizes from traditional distributed learning without attacks, our theoretical results point out the improper setting in most of the previous works.
> > >
> > > 2. We further improve the upper bound for ByzSGDm. Since the order of $O(1/T^\frac{1}{4})$ has been proven to be optimal for SGD (Arjevani et al., 2019), we believe that our improvement is meaningful.
> > >
> > > 3. Moreover, we introduce the normalized momentum and propose ByzSGDnm mainly to improve the empirical performance, as in many previous works on large-batch training (Goyal et al., 2017; Hoffer et al., 2017; Keskar et al., 2017; You et al., 2020). ByzSGDnm with a relatively large batch size significantly outperforms the baselines in our experiments, and has a similar theoretical convergence rate as ByzSGDm (which has been proven to be optimal).
> > >
> > > Given the reasons above, we believe there are both adequate theoretical contributions and empirical contributions in our work. We would be sincerely grateful if the reviewer could re-evaluate the contribution of our work in light of our response.

---

### Official Review · Reviewer_Rysg · 2023-11-01

**Soundness:** 3 good
**Presentation:** 3 good
**Contribution:** 3 good
**Rating:** 6
**Confidence:** 3

**Summary:**

In this paper, the authors study the optimization problem of Byzantine-robust distributed learning in an i.i.d. case. They propose a new method, called Byzantine-robust stochastic gradient descent with normalized momentum. They prove the convergence guarantee for this algorithm and theoretically analyze the optimal value of batch size.  Also, the dependence of the rate on the batch size is studied experimentally.

**Strengths:**

1. The presentation of results is written well.
2. Good deep learning experiments.

**Weaknesses:**

1. In the experimental part of the work, the experiments do not support theoretical analysis. The authors do not compare the performance of the method with optimal batch size value and with another possible choice of it. It would be better if such experiments were in the work.
2. Also, the experiments with the comparison of convergence rates of the proposed method and previous methods. It would be better to provide some experiments like in this paper https://arxiv.org/pdf/2206.00529.pdf (see Figure 1).
3. In the work the authors consider a homogeneous setting. Some results in heterogeneous setup can improve the contribution of this work dramatically.

**Questions:**

1. There is no theoretical comparison between ByzSGDnm and ByzSGDm. Are there any theoretical benefits of ByzSGDnm compared to ByzSGDm?

 Please see the weaknesses.

---

> ### Author Response · Authors · 2023-11-21
>
> We sincerely thank the reviewer for the valuable comments. We have revised our work according to the suggestions and updated the revised version. The changed or added parts are marked in blue for quick recognition. Then, we respond to the raised concerns and questions point by point below.
>
> ---
>
> **Weakness 1. The authors do not compare the performance of the method with optimal batch size value and with another possible choice of it.**
>
> Since the constants $L$ and $F_0$ for the ResNet-20 deep learning model on the CIFAR-10 dataset is hard to obtain, we try different batch sizes ranging from $2^5\times 8$ to $2^{10}\times 8$ in the experiment. As the empirical results in Table 1 and Table 2 show, for ByzSGDm with each of the four aggregators (KR, GM, CM, and CC), the optimal batch size $B^*$ that leads to the best top-$1$ test accuracy increases as $\delta$ increases from $0$ to $3/8$. Moreover, the top-$1$ accuracy increases as batch size $B$ increases when $B<B^*$, and decreases as $B$ increases when $B>B^*$. The empirical results are highly consistent with our theoretical results.
>
> Similar empirical results for ByzSGDnm can be found in Table 7, Table 8, Table 9, Table 10, and Table 11 in Appendix E.1 due to limited space in the main text.
>
> ---
>
> **Weakness 2. It would be better to provide some experiments with the comparison of convergence rates of the proposed method and previous methods.**
>
> We sincerely thank the reviewer for the valuable suggestion. Due to limited space in the main text, we have added more results in the Appendix. Please see Figure 1 to Figure 9 in Appendix D.5 and Appendix E, which illustrate the empirical convergence rate of our method and the existing methods under different settings. The added results in the figures are highly consistent with those in the initial version.
>
> ---
>
> **Weakness 3. Some results in heterogeneous setup can improve the contribution of this work dramatically.**
>
> We thank the reviewer for the constructive suggestion. We have added the empirical results in heterogeneous setup in Appendix E.2. For quick access, we briefly summarize the empirical results below.
>
> We sample from the training set of CIFAR-10 using the Dirichlet distribution with hyper-parameter $1.0$ to create the heterogeneous data. Among the $8$ workers, $1$ worker is under ALIE attack. Geometric median is used as the aggregator. The final top-$1$ test accuracy for different methods is presented in the table below.
>
> |Batch size|32$\times$8 | 64$\times$8 | 128$\times$8 | 256$\times$8 | 512$\times$8|
> |----|----|----|----|----|----|
> |Byz-VR-MARINA $(p=0.05)$ | 23.07\% | 25.01\% | 22.25\% | 24.67\% |14.28\% |
> |Byz-VR-MARINA $(p=0.1)$ | 27.16\% | 23.19\% | 23.08\% | 13.86\% | 11.45\% |
> |Byz-VR-MARINA $(p=0.2)$ | 17.83\% | 28.10\% | 27.34\% | 18.67\% | 25.21\% |
> |ByzSGDm | 10.16\% | 23.33\% | 27.78\% | 29.63\% | 32.90\% |
> |ByzSGDnm | 23.87\% | 26.78\% | 28.42\% | 30.04\% | **35.24\%**|
>
> As the empirical results show, under this non-i.i.d. setting, using a relatively large batch size can still lead to higher test accuracy. Moreover, ByzSGDnm with batch size $512\times 8$ has the best final top-$1$ test accuracy.
>
> In addition, we have also tested the empirical performance of the three methods (ByzSGDnm, ByzSGDm, Byz-VR-MARINA) when combined with nearest neighbor mixing (NNM) technique under the non-i.i.d. setting. ByzSGDnm still has the best final top-$1$ test accuracy among the methods. Please refer to Appendix E.2. for more detailed results.
>
> ---
> **Question 1. There is no theoretical comparison between ByzSGDnm and ByzSGDm. Are there any theoretical benefits of ByzSGDnm compared to ByzSGDm?**
>
> To the best of our knowledge, the theoretical upper bounds for methods based on SGD (without normalization) in non-convex cases are usually in the form of "$\mathbb{E}||\nabla F(w_t)||^2\leq ...$" while the upper bounds for those based on normalized SGD in the form of "$\mathbb{E}||\nabla F(w_t)||\leq...$". Therefore, the theoretical upper bound for ByzSGDnm cannot be directly compared to ByzSGDm. As we present in Section 4, the obtained convergence order $O(1/T^\frac{1}{4})$ for ByzSGDnm is optimal. Moreover, ByzSGDnm can recover the convergence rate of normalized momentum SGD when there is no attack ($\delta=0$).
>
> Methods based on SGD (without normalization) and those based on normalized SGD have their own applications. Methods based on normalized SGD have been reported to have better empirical performance in large-batch cases (Goyal et al., 2017; Hoffer et al.,2017; Keskar et al., 2017; You et al., 2020). Given the findings in previous works above, we propose ByzSGDnm since our theoretical results show that a relatively large batch size is preferred under Byzantine attacks.
>
> ---
> We hope that we have addressed the reviewer's concerns, and we are always willing to respond to any further concerns. Meanwhile, we would greatly appreciate it if the reviewer could re-evaluate our work in light of our response.

---

> > ### Comment · Reviewer_Rysg · 2023-11-22
> >
> > Thank you for your response!
> >
> > The new added experiments are good! However, the experiments, which I have expected to see, might be in the same manner as Figure 1 from paper  https://arxiv.org/pdf/2206.00529.pdf. What I mean is that to support your theoretical results it would be better to compare methods from your experiments but in another metric:  how $||\nabla F(w_t)||_2$ or $||\nabla F(w_t)||^2_2$ decrease during iterations.

---

> > > ### Author Response · Authors · 2023-11-22
> > >
> > > We thank the reviewer for the timely response and the positive comments on our added experiments. We are sorry for misunderstanding the reviewer's initial comments. Meanwhile, we would like to point out that in non-i.i.d. cases, a point with small $||\nabla F(w_t)||$ can have a large loss $F(w_t)$. Therefore, $||\nabla F(w_t)||$ w.r.t. iteration number may not properly reflect the performance of a training method. Due to this reason, we did not record $||\nabla F(w_t)||$ in our experiments before, and the remaining time is not enough for us to re-run the experiments.
> > >
> > > Instead, to address the reviewer's concern, we added results of training loss $F(w_t)$ w.r.t. gradient computation number (or, equivalently, epoch number) in the second revision. Please refer to Figure 8 in Appendix E.1 on page 32 in the latest version. The added results further verify the effectiveness of large batch size and ByzSGDnm.
> > >
> > > Meanwhile, we would like to point out that as far as we know, even for non-convex objective functions, $F(w)$ w.r.t. iteration is much more frequently used as the metric in previous works than $||\nabla F(w_t)||$. Here are some references:
> > >
> > > + Figure 4-5 from (Xie et al., 2020);
> > > + Figure 5-7 in Appendix C from (Yang and Li, 2021);
> > > + Figure 6 in Appendix A.2.3 from (Karimireddy et al., 2022);
> > >
> > > We sincerely thank the reviewer for the response again and hope that our response can address the reviewer's concern. Meanwhile, we will take the reviewer's suggestion into consideration and add results about $||\nabla F(w_t)||$ in future versions. We would greatly appreciate it if the reviewer could re-evaluate our work in light of our response.
> > >
> > > ---
> > >
> > > Cong Xie, Oluwasanmi Koyejo, and Indranil Gupta. Fall of empires: Breaking Byzantine-tolerant sgd by inner product manipulation. UAI 2020.
> > >
> > > Yi-Rui Yang and Wu-Jun Li. BASGD: Buffered asynchronous sgd for Byzantine learning. ICML 2021.
> > >
> > > Sai Praneeth Karimireddy, Lie He, and Martin Jaggi. Byzantine-robust learning on heterogeneous datasets via bucketing. ICLR 2022.

---

### Official Review · Reviewer_jRhg · 2023-11-01

**Soundness:** 3 good
**Presentation:** 3 good
**Contribution:** 3 good
**Rating:** 6
**Confidence:** 3

**Summary:**

In this paper, the authors show the effect of batch size on the performance of robust algorithm against Byzantine attacks. More specifically, they characterize the optimal batch size $B^\star$ to choose when the number of gradient computations is fixed. In addition, they present ByzSGDnm, a robust algorithm that uses stochastic gradient descent with normalized momentum. The authors provide a theoretical guarantee of the algorithm on non-convex functions and provide empirical results showing the efficiency of the proposed method.

**Strengths:**

- The paper is clear and easy to follow.
- Increasing the batch size and normalizing the gradient significantly improve the empirical performance of the model in the i.i.d case.

**Weaknesses:**

- The proposed algorithm is only studied in a homogeneous setting ("i.i.d. case" in the paper), which is generally not the case in real applications where data between different clients are heterogeneous. Does the author have any ideas or perhaps experimental results on the behavior of the proposed algorithm in the presence of heterogeneous data?

**Questions:**

Table 6 shows the execution time of the different algorithms for different batch sizes, for a specific and fixed number of epochs. Can the authors explain why they chose to show the results for a fixed number of epochs and not for a specific accuracy achieved? As far as I know, the speed gain obtained by choosing a larger batch size is naturally explained by the fact that we can benefit from the parallelization of computations (in the system side). However, I would find it interesting to understand whether the methods presented are faster to reach a given accuracy with a larger batch size.

---

> ### Author Response · Authors · 2023-11-21
>
> We sincerely thank the reviewer for the constructive comments and the support of our work. We have revised our work according to the suggestions and updated the revised version. The changed or added parts are marked in blue for quick recognition. Then, we respond to the raised concerns and questions point by point below.
>
> **Weakness 1: Does the author have any ideas or perhaps experimental results on the behavior of the proposed algorithm in the presence of heterogeneous data?**
>
> We thank the reviewer for the constructive suggestion. We have added the empirical results in the presence of heterogeneous data in Appendix E.2. For quick access, we briefly summarize the empirical results below.
>
> We sample from the training set of CIFAR-10 using the Dirichlet distribution with hyper-parameter $1.0$ to create the heterogeneous data. Among the $8$ workers, $1$ worker is under ALIE attack. Geometric median is used as the aggregator. The final top-$1$ test accuracy for different methods is presented in the table below.
>
> |Batch size|32$\times$8 | 64$\times$8 | 128$\times$8 | 256$\times$8 | 512$\times$8|
> |----|----|----|----|----|----|
> |Byz-VR-MARINA $(p=0.05)$ | 23.07\% | 25.01\% | 22.25\% | 24.67\% |14.28\% |
> |Byz-VR-MARINA $(p=0.1)$ | 27.16\% | 23.19\% | 23.08\% | 13.86\% | 11.45\% |
> |Byz-VR-MARINA $(p=0.2)$ | 17.83\% | 28.10\% | 27.34\% | 18.67\% | 25.21\% |
> |ByzSGDm | 10.16\% | 23.33\% | 27.78\% | 29.63\% | 32.90\% |
> |ByzSGDnm | 23.87\% | 26.78\% | 28.42\% | 30.04\% | **35.24\%**|
>
> As the empirical results show, under this non-i.i.d. setting, using a relatively large batch size can still lead to higher test accuracy. Moreover, ByzSGDnm with batch size $512\times 8$ has the best final top-$1$ test accuracy.
>
> In addition, we have also tested the empirical performance of the three methods (ByzSGDnm, ByzSGDm, Byz-VR-MARINA) when combined with nearest neighbor mixing (NNM) technique under the non-i.i.d. setting. ByzSGDnm still has the best final top-$1$ test accuracy among the methods. Please refer to Appendix E.2. for more detailed results.
>
> **Question 1: Can the authors explain why they chose to show the results for a fixed number of epochs and not for a specific accuracy achieved?**
>
> We show the results for a fixed number of epochs in order to support the claim that using a relatively large batch size can more efficiently utilize the computation power of GPUs. In other words, we want to empirically verify that using a relatively large batch size can make more gradient computation finished in a unit of time.
>
> We have followed the reviewer's suggestion and presented the wall-clock running time that different methods require to reach 50\%, 75\%, and 85\% top-$1$ test accuracy in Appendix E.1. The results show that ByzSGDnm requires less time to reach the target accuracy than ByzSGDm in most cases. Meanwhile, using a relatively large batch size can make ByzSGDm and ByzSGDnm reach the given accuracy faster. For quick access, we present the wall-clock running time required to reach 75\% top-$1$ test accuracy below. The missing values mean that the target test accuracy is not reached in $160$ epochs for the corresponding methods.
>
> |Batch size|32$\times$8 | 64$\times$8 | 128$\times$8 | 256$\times$8 | 512$\times$8|1024$\times$8|
> |----|----|----|----|----|----|----|
> |ByzSGDm + KR | -- | -- | -- | -- | **133.69s** | 148.70s|
> |ByzSGDnm + KR | -- | -- | 328.49s | **123.12s** | 130.45s | 134.09s|
> |ByzSGDm + GM | -- | -- | 114.38s | **59.19s** | 90.74s | 124.03s|
> |ByzSGDnm + GM | -- | 252.19s | 66.13s | 54.12s | **45.07s** | 76.49s|
> |ByzSGDm + CM | -- | -- | -- | 202.97s | **110.25s** | 204.42s|
> |ByzSGDnm + CM | -- | -- | 322.83s | 150.25s | **63.85s** | 109.23s|
> |ByzSGDm + CC | -- | 727.07s | 77.19s | 87.42s | **86.53s** | 150.59s|
> |ByzSGDnm + CC | 1465.07s | 112.49s | 109.18s | **56.20s** | 61.30s | 81.28s|
>
> ---
>
> We sincerely thank the reviewer for their valuable time and their support of our work again. Meanwhile, we would greatly appreciate it if the reviewer could re-evaluate our work in light of our response.

---

### Meta-Review · Area_Chair_BgGn · 2023-12-09

**Metareview:**

While the scores are borderline, all the reviews are positive, and reviewers appreciate the new experiments done in response to the comments. Overall the paper makes a solid contribution in the literature of distributed learning.

**Justification For Why Not Higher Score:**

There was concern regarding theoretical improvement of the paper compared to existing BDRL methods.

**Justification For Why Not Lower Score:**

The authors provided new experiments to show the effectiveness of their paper.

---

### Decision · Program_Chairs · 2024-01-16

Accept (poster)